# TRIGO: Benchmarking Formal Mathematical Proof Reduction for Generative Language Models

**Jing Xiong**[1*], **Jianhao Shen**[2*], **Ye Yuan**[2], **Haiming Wang**[5], **Yichun Yin**[6],
**Zhengying Liu**[6], **Lin Li**[6], **Zhijiang Guo**[6], **Qingxing Cao**[1], **Yinya Huang**[1,4],
**Chuanyang Zheng**[3], **Xiaodan Liang**[1†], **Ming Zhang**[2†], **Qun Liu**[6]

[1]Shenzhen Campus of Sun Yat-Sen University    [2]Peking University
[3]The Chinese University of Hong Kong    [4]City University of Hong Kong
[5]Sun Yat-Sen University    [6]Huawei Noah's Ark Lab

{xiongj69, wanghm39, caoqx}@mail2.sysu.edu.cn,
{jhshen, yuanye_pku, mzhang}@pku.edu.cn,
{yinyichun, liuzhengying2, guozhijiang, lilin29, qun.liu}@huawei.com
{cyzheng21}@cse.cuhk.edu.hk {yinya.el.huang, xdliang328}@gmail.com

## Abstract

Automated theorem proving (ATP) has become an appealing domain for exploring the reasoning ability of the recent successful generative language models. However, current ATP benchmarks mainly focus on symbolic inference, but rarely involve the understanding of complex number combination reasoning. In this work, we propose TRIGO, an ATP benchmark that not only requires a model to reduce a trigonometric expression with step-by-step proofs but also evaluates a generative LM's reasoning ability on formulas and its capability to manipulate, group, and factor number terms. We gather trigonometric expressions and their reduced forms from the web, annotate the simplification process manually, and translate it into the "Lean" formal language system. We then automatically generate additional examples from the annotated samples to expand the dataset. Furthermore, we develop an automatic generator based on Lean-Gym to create dataset splits of varying difficulties and distributions in order to thoroughly analyze the model's generalization ability. Our extensive experiments show our proposed TRIGO poses a new challenge for advanced generative LM's including GPT-4 which is pre-trained on a considerable amount of open-source formal theorem-proving language data, and provide a new tool to study the generative LM's ability on both formal and mathematical reasoning.

## 1 Introduction

Automated theorem proving (ATP) requires formal reasoning and deduction from conclusion to axioms or known theorems. This task requires general and

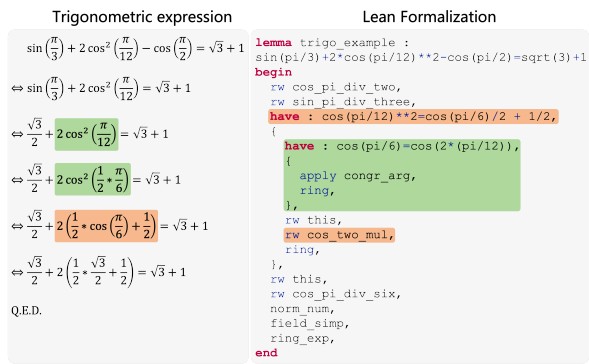

Figure 1: The task of trigonometric expression reduction. The key is to rewrite $\frac{\pi}{12}$ into $\frac{1}{2} * \frac{\pi}{6}$ (the green part), and apply the half-angle formula (the orange part). Both steps need an understanding of numbers and formulas.

flexible reasoning and is easy to validate, making it an appealing domain for exploring the reasoning ability of the recent successful pre-trained generative language models. These models show strong proof generation capabilities (Lample et al., 2022; Jiang et al., 2022), but its ability to perform formal mathematical proof reduction, which involves complex numerical reasoning, has not been thoroughly explored.

Current ATP benchmarks (Wu et al., 2021a; Han et al., 2021; Zheng et al., 2022) mainly focus on symbolic inference but rarely involve the understanding of complex number combination reasoning, such as term grouping, term factorization and equivalent substitution. For advanced mathematical proving such as trigonometric expressions, ATP can be beneficial for evaluating the crucial complex number combination. For example, as shown in Figure 1, to correctly reduce the left-hand-side expression, one must recognize the specific angles or terms such as $\cos(\frac{\pi}{12})$, capable of applying the half-angle formula and know the result is $\cos(\frac{1}{2} * \frac{\pi}{6})$.

---

*These authors contributed equally.
†Corresponding author

Figure 2: An example of GPT-4 struggling to solve TRIGO. GPT-4 corrects its response after the second prompt (box in right), but continues generating error tactics (highlighted in red with a cross mark).

Such proof steps can be automatically denoted by the formal language deduction as shown in the right-hand-side, and then verified via the interactive theorem-proving environment in ATP.

To develop such profound ATP evaluation for current generative LMs, we propose the task of Trigonometric Expression Reduction (TRIGO). Given a trigonometric expression, a model is required to accept formal input with Lean formal language and then perform step-by-step proof reduction. The proposed TRIGO poses a new challenge for current state-of-the-art generative LMs. Figure 2 presents an illustrative example of a proof generated by GPT-4 (OpenAI, 2023). Inspired by self-refine (Madaan et al., 2023), we first provide the prompt "Please help me prove this problem using Lean: lemma Trigo_0 : sin(107 * pi) = 0 :" to GPT-4 and the GPT-4 generates non-exist tactics and gives the incorrect equations. We further prompt it with "We require to use lemma from Lean's standard library for the proof and to ensure the correctness of the equation" to correct the GPT-4. However, in the second attempt, the GPT-4 still applies the tactic "sin_periodic_pi" that does not exist in the Lean standard library and comments it "−−available in Lean standard library". Surprisingly, GPT-4 produced the correct equation "have h: sin(107*pi) = sin(1*pi)" in the second proof attempt, even though the proof for this subgoal is incorrect. This example demonstrates the potential of GPT-4 in accurately manipulating numbers and formulas, as well as the challenge of strict formal reasoning posed by the TRIGO task.

To construct the TRIGO dataset, we collect trigonometric expression reduction problems and corresponding answers from high school exercises and exams. We then develop an interactive annotation software to manually label the reduction steps and formalize the processes into "Lean" formal language. Finally, based on this manually formalized data, we develop an automatic proof generation program to expand the dataset with real-world data and create datasets of artificially generated samples. Specifically, we generate 3 types of samples by controlling their proof length and generating trigonometric functions with larger numerical values to assess the models' ability to generalize to out-of-distribution data.

Our contributions are three-fold:
- We propose the new trigonometric expression reduction tasks that are the first to explore formal mathematical reasoning abilities with regard to both formulas and numerical elements understanding.
- We construct the new TRIGO dataset with manually labeled reduction steps and convert them to the formal language Lean (de Moura et al., 2015). We also generate extra samples with controlled difficulties and distribution to further evaluate different aspects of generative LMs.
- We conduct extensive experiments and detailed analysis of a broad range of methods, identifying the new challenges for current state-of-the-art generative LMs.

## 2 Related Work

Automatic theorem proving (ATP) has numerous formal environments such as HOList (Bansal et al., 2019a), Metamath (Megill and Wheeler, 2019), and CoqGYM (Yang and Deng, 2019). There are several formal benchmark tasks for theorem proving that exist. LeanStep (Han et al., 2021) extracts (state, tactic) pairs from mathlib (mathlib, 2020), a

comprehensive library of theorem proofs in Lean. The IsarStep dataset (Li et al., 2021) mines intermediate proof steps from the Archive of Formal Proofs (AFP). MiniF2F (Zheng et al., 2022) is a cross-system benchmark of olympiad-level mathematics problems, containing 488 problems formalized in Metamath, Lean, Isabelle, and HOL Light, but only a small portion has formalized solutions. FIMO (Liu et al., 2023) targets formal International Mathematical Olympiad (IMO) problems. The Geometry3K dataset (Lu et al., 2021) includes 3,002 geometry problems with annotated formal language descriptions but lacks interaction with formal environments. Many other works also focus on informal math problem solving (Saxton et al., 2019; Hendrycks et al., 2021; Cobbe et al., 2021; Shen et al., 2021). Our work is most similar to (Wu et al., 2021a). They use formal mathematical reasoning to reduce equations and inequalities and employ programs to automatically generate proofs to explore combinational generalization. However, they lack real-world problems to assess the model's generalization to real distributions and do not involve complex numerical operations combination. Compared with previous work (Wu et al., 2021a) which has 18 axioms and 9 transformations, our generation process has a total of 85 transformation rules and diverse sampled parameters. Our proposed TRIGO generates complex samples with controlled difficulties and distribution, and includes manually annotated samples and proof steps from real-world problems.

## 3 Background on Lean Environment

Formal language systems are effective tools for strictly verifying the correctness of each proof step generated by the model. In this work, we use Lean (de Moura et al., 2015) as formal environment.

The correctness of a given proof can be verified by a Lean verifier program. A Lean proof example is shown in Figure 1. It starts from a **goal state** "$\vdash \sin(\pi/3) + 2 * \cos(\pi/12) * * 2 - \cos(\pi/2) = \text{sqrt}(3) + 1$", representing the current proof goal. In the row next to the "begin", a **tactic** "rw cos_pi_div_two" is applied to the current goal state, meaning rewrites the term $\cos(\pi/2)$ to 0. In the following rows, the proof applies "have" tactic generates a new sub-goal such as proving $\cos(\pi/12)^2 = \cos(\pi/6)/2 + 1/2$, and applies the "ring_exp" to solving exponents equations in com-

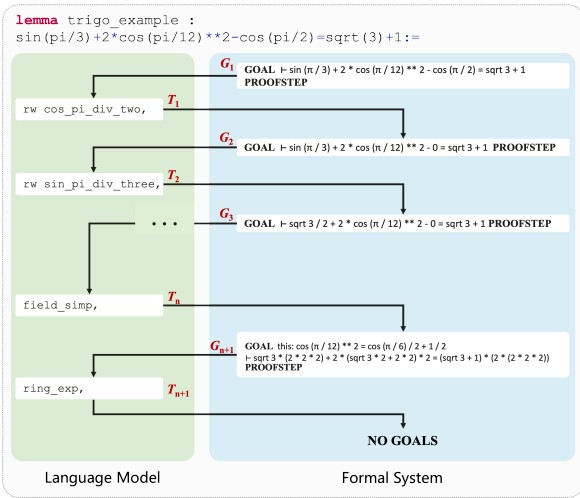

Figure 3: The proof flow is produced by the interactive Lean-Gym environment. The language model generates proof steps given the formal prompts until reaches "no goal".

mutative (semi)rings. More details of used tactics are given in Appendix H.

Lean-Gym (Polu et al., 2023) is an interactive environment that allows language models to interact with formal systems. As depicted in Figure 3, we begin by acquiring the initial **goal state** $G_1$ as "$\vdash \sin(\pi/3) + 2 * \cos(\pi/12) * * 2 - \cos(\pi/2) = \text{sqrt}(3) + 1$". This goal state is inputted into language model with the prompt "GOAL $G_1$ PROOFSTEP". Subsequently, GPT-2 generates the corresponding **tactic** $T_1$ as "rw cos_pi_div_two,". Given the **goal state** and **tactic**, Lean-Gym outputs a new **goal state** "$\vdash \sin(\pi/3) + 2 * \cos(\pi/12) * * 2 - 0 = \text{sqrt}(3) + 1$" for language model to obtain the next **tactic**. We iteratively perform this process until the Lean-Gym returns "**no goals**" which indicates the proof is complete.

## 4 TRIGO Dataset

In this section, we first introduce how we collect trigonometric expression reduction problems from "tiku"[1], annotate the step-by-step reduction processes, and transform them into Lean formal language to create the TRIGO-real and TRIGO-web datasets. Then we introduce how to automatically generate data to construct the TRIGO-gen.

### 4.1 Problem Collection

We collect the trigonometric expression reduction problems from "tiku", a large-scale math problem set from textbooks and exams. Specifically, we collect problems and their answers from

---

[1] https://www.tiku.cn/

the "trigonometry" topic. We eventually collect 427 problems and denote them as TRIGO-real. To expand our dataset, we further collect additional trigonometry reduction problems from different websites. After manually filtering the duplicate problems, we obtain an additional 453 samples as TRIGO-web and use them as the test set. These data are collected from other websites found through search engines. These sources contain high school math exam questions with standard answers. Throughout the collection process, we aim to gather data randomly whenever possible, ensuring diversity in the distribution of the test set to reflect the model's performance on real human exam questions.

### 4.2 Interactive Proof Annotation

The collected problems have only the final results without step-by-step reduction processes. To facilitate the annotation of these crucial processes, we develop interactive software specifically tailored for this purpose. The annotation process has the following steps:

Step. 1 The software shows an expression to the annotator.

Step. 2 The annotator inputs a transformation equation that will be applied in the next step.

Step. 3 The software checks if the equation is valid by matching it with a rule in a predefined bank. If no rule is matched, the software reports "No Matched Rule" and goes back to step 2.

Step. 4 The software applies this transformation to the current problem. If succeeds, the software outputs the new expression. Otherwise the software reports "Rule Failed" and goes back to step 2.

Step. 5 Repeat steps 2-4 until the expression equals the answer.

**Equation-Rule Matching** In step 3, each annotated equation must match with a predefined rule to ensure its correctness. We define a total of 85 rules that can cover most of the trigonometric transformation. Some examples are shown in Table 1. As shown in Table 1, some rules have value argument $X$, $Y$, and parameter $K$. To perform the rule matching, we first use sympy[2] to parse both

[2]https://docs.sympy.org/latest/tutorial/manipulation.html

| Identity Rule Name | Example |
|---|---|
| sin_zero | $\sin(0) = 0$ |
| sin_pi_div_six | $\sin\left(\frac{\pi}{6}\right) = \frac{1}{2}$ |
| cos_pi | $\cos(\pi) = -1$ |
| cos_neg | $\cos(X) = \cos(2 * \pi * K - X)$ |
| tan_add | $\tan(X + Y) = \frac{\tan(X) + \tan(Y)}{1 - \tan(X) * \tan(Y)}$ |

Table 1: Examples of our pre-defined rule bank.

the equation and rules into expression trees. Then we compare each equation sub-tree with the rule tree. If two trees are identical except for the sibling nodes' order, we consider they are matched and the equation is valid. Given a matched sub-tree, we further use sympy to extract the arguments $X$, $Y$, and $K$. The full list of examples is demonstrated in Appendix I.

### 4.3 Lean Formalization

After obtaining the stepwise reduction annotation, we manually transform the annotated equation into Lean formal language. Since we use Lean-Gym with mathlib (mathlib, 2020) backend as our formal environment, Lean-Gym can only accept tactics inside the mathlib. To ensure the correct acceptance and processing of our defined trigonometric rules in Lean-Gym, we derive these rules from mathlib theorems and convert them into tactics before adding them to mathlib.

We then construct the framework of the proof script. The script begins with the keyword "lemma" followed by a name and the premises of the lemma and then presents the goal equation where the left-hand side (LHS) represents the original expression and the right-hand side (RHS) represents its reduced result. Lastly, we add the "begin", "sorry", and "end" keywords where the "sorry" is a placeholder that will be replaced in the following steps.

Given the empty proof script, we convert the annotated step into Lean tactics. Recall that during the annotation phase, each annotated step is matched with a predefined rule, which can be further converted to a Lean-Gym tactic using a Python program. Thus, we only need to apply the corresponding rule with proper arguments parsed by sympy. Take the equation $\sin\left(\frac{13\pi}{6}\right) = \sin\left(\frac{\pi}{6}\right)$ as an example. The matched rule is $\sin(X) = \sin(X - 2\pi)$ with argument $X = \frac{13\pi}{6}$. After applying this rule to sympy we get a new goal equation: $\sin\left(\frac{13\pi}{6} - 2\pi\right) = \sin\left(\frac{\pi}{6}\right)$, along with the corresponding tactics.

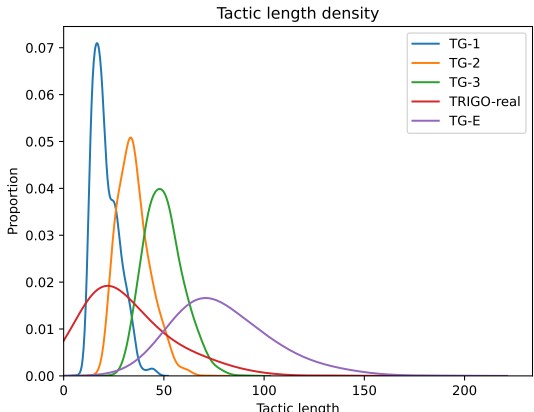

Figure 4: Tactic length distribution based on the number of tactics.

Although the above step can complete most of the transformation to Lean, we still need to manually fix the Lean proof. For example, Lean does not reduce the above new goal state $\sin\left(\frac{13\pi}{6} - 2\pi\right)$ to $\sin\left(\frac{\pi}{6}\right)$, and rewriting tactics "rw $\sin(x+y) = \cdots$" fails when applied to "$\sin(y + x)\cdots = \cdots$" as Lean can not match $\sin(x + y)$ with $\sin(y + x)$. Thus, we manually add more steps such that the Lean-Gym can correctly process the entire proof.

### 4.4 Generated Data

To comprehensively analyze the performance of models across various levels of difficulty and different ranges of numbers, as well as to study the gap between generated and real-world data, we automatically generate trigonometric problems and proofs by applying random predefined rules repeatedly. Specifically, we randomly choose a rule denoted as $r$ from our predefined rule bank and select corresponding value arguments $X$ and $Y$ from the value list $C = \{2\pi, \pi, \frac{\pi}{2}, \frac{\pi}{3}, \frac{\pi}{6}, -\frac{\pi}{6}, -\frac{\pi}{3}, -\frac{\pi}{2}, -\pi, -2\pi\}$, and initialize $K$ with an integer value between 0 and 100, to construct our initial goal expressions $G$. Then at each step, we sample a $r$ and try to match the $r$ with LHS or RHS of goal expressions $G$. If either side is matched, we replace the corresponding part of goal expressions $G$ with rule $r$'s corresponding Lean tactics set.

During the replacement, we need to determine the value arguments $X, Y$, and $K$ given the goal $G$. For example, consider the expression $\sin(\frac{3\pi}{4})$ and the rule $\sin(X + Y) = \sin(X)\cos(Y) + \sin(Y)\cos(X)$, where the argument $X + Y$ must equal $\frac{3\pi}{4}$. We first sample the value of $X$ from the value list $C$, then calculate the $Y = \frac{3\pi}{4} - X$.

To obtain the parameter $K$ for some rules such as $\cos(X) = \sin(2 * \pi * K - X + \frac{\pi}{2})$, we uniformly choose an integer between $[0, 100]$ as its value.

To control the difficulties of the generated samples, under the assumption that the difficulty of the problem increases as the number of sampled rules grows, we sample and apply 1, 2, and 3 rules to construct TRIGO-gen, denoted as TG-1, TG-2, and TG-3, respectively. For each rule length, we generate 9,000 training samples, 1,000 validation samples, and 1,000 test samples. To close the gap between the generated samples and real-world samples, we also use the trigonometric expression in TRIGO-real as initial goal expressions $G$, then sample and apply the exact 3 rules to generate the set TG-E as generated training data.

### 4.5 Data Statistics

Finally, the TRIGO-real has 427 problems and a total of 10,574 proof tactics. We divide TRIGO-real into train, validation, and test splits with a 7:1:2 ratio, resulting in 299 training samples, 42 validation samples, and 86 test samples. The average proof step size for TRIGO-real, TG-1, TG-2, TG-3, and TG-E are 37, 22, 35, 49 and 81. Since Lean-Gym only accepts one tactic at a time, the tactic length of each problem in the dataset typically matches the size of the proof step. Figure 4 displays the sample proportions concerning their tactic length. We can observe that the generated samples have similar tactic lengths, while the real-world data has various but more uniform lengths. More statistics are in Appendix G.

## 5 Baseline Models

Recent works utilize GPT-based language models for automated theorem proving and have made significant improvement (Polu and Sutskever, 2020; Han et al., 2021; Polu et al., 2023; Jiang et al., 2022; Zheng et al., 2023). In this work, we use GPT-2 (Radford et al., 2019) with a proof search algorithm as a baseline method for our dataset.

**Data Preparation** The Lean-Gym (Polu et al., 2023) provides an interactive formal environment to obtain a new goal state given the previous state and tactic, as shown in Figure 3. During training, at each step, we obtain the (state, tactic) pairs from Lean-Gym and training samples respectively, and concatenate them into a sequence with the "GOAL" and "PROOFSTEP" special tokens:

GOAL ⟨state⟩ PROOFSTEP ⟨tactic⟩.

| Model | Manual Labeling | | TRIGO-gen | | |
|---|---|---|---|---|---|
| | TRIGO-real | TRIGO-web | TG-1 | TG-2 | TG-3 |
| GPT-2$_B$ | 12.79 | 13.90 | 42.79 | 7.49 | 0.39 |
| GPT-2$_L$ | 12.79 | 13.02 | 71.59 | 23.39 | 1.69 |
| GPT-2$_L$-PACT | 32.55 | 25.60 | 77.29 | 44.19 | 18.99 |
| GPT-2$_{B-D}$ | 17.44 | 17.21 | 42.29 | 12.69 | 4.89 |
| GPT-2$_{L-D}$ | 19.76 | 20.08 | 54.79 | 20.89 | 7.69 |
| GPT-2$_L$-PACT-D | 23.25 | 13.02 | **84.29** | **60.09** | **25.29** |
| GPT-2$_L$-PACT-E | **34.88** | **25.38** | - | - | - |

Table 2: Pass rates of benchmark models and baselines.

We take the above sequence as input and train the GPT-2 models to predict the "tactic" sequence with autoregressive loss (Bengio et al., 2000):

$$\mathcal{L}\left(\theta\right) = -\sum_{i=1}^{n-1} \log p\left(x_{i+1}|x_1, x_2, ..., x_i; \theta\right),$$

where $\theta$ indicates model parameters, and $x_i$ is the $i$-th token of the input sequence:

**Proof Search** After training GPT-2 to generate a tactic given a goal state, we search the complete proof by expanding the most probable state at each step. We employ Breadth-First Search (BFS) in this paper. Specifically, we define the probability of the goal state as the cumulative logarithm probability of its corresponding generated tactics:

$$\log P_{state_N} = \log p_{state_{N-1}} + \log p_{tactics_i}, \quad (1)$$

where $p_{tactics_i}$ is the tactic's probability generated by the GPT-2. Lean-Gym outputs the new state $state_N$ by applying the $tactics_i$ to a previous goal state $state_{N-1}$. At each proof search step, we select a goal state with the highest probability and feed the sequence "GOAL ⟨state⟩ PROOFSTEP" into the trained GPT-2 to generate the tactics. We sample 8 tactics based on GPT-2 output probability. The generated tactics with the goal state are input to Lean-Gym to obtain a new valid goal state if possible. We repeat the search process until "**no goal**" state is reached, the queue becomes empty, or reach the maximum search step 512.

# 6 Experiment

In this section, we evaluate the performance of GPT-2$_{BASE}$ (GPT-2$_B$), GPT-2$_{LARGE}$ (GPT-2$_L$), and GPT-2$_L$-PACT, a GPT-2$_{LARGE}$ pre-trained on the formal proof dataset PACT (Han et al., 2021). Furthermore, we evaluate the model's out-of-distribution generalization ability by examining its performance across various levels of difficulty and different ranges of numbers, while also evaluating

| Model | Manual Labeling | | TRIGO-gen | | |
|---|---|---|---|---|---|
| | TRIGO-real | TRIGO-web | TG-1 | TG-2 | TG-3 |
| *Training on TG-1* | | | | | |
| GPT-2$_B$ | **9.30** | **0.94** | - | 9.89 | 3.39 |
| GPT-2$_L$ | 5.81 | 0.88 | - | 14.29 | 6.19 |
| GPT-2$_L$-PACT | 5.81 | 0.88 | - | **16.39** | **6.29** |
| *Training on TG-2* | | | | | |
| GPT-2$_B$ | 3.48 | 0.44 | 19.59 | - | 2.59 |
| GPT-2$_L$ | 4.65 | 0.66 | 32.29 | - | 8.29 |
| GPT-2$_L$-PACT | **5.81** | **1.32** | **48.49** | - | **15.89** |
| *Training on TG-3* | | | | | |
| GPT-2$_B$ | 1.16 | 0.00 | 5.59 | 0.89 | - |
| GPT-2$_L$ | 0.00 | 0.00 | 10.59 | 3.49 | - |
| GPT-2$_L$-PACT | **8.13** | **1.10** | **57.19** | **46.39** | - |

Table 3: Pass rates of models on OOD test set.

the impact of generating data distributions beyond those observed in real-world data. Additionally, we conduct a comprehensive analysis of the models, including an evaluation of GPT-4's performance.

## 6.1 Implementation Details

All models are trained with Adam optimizer (Kingma and Ba, 2015), learning rate of $2.5 \times 10^{-4}$, batch size of 512, and a cosine schedule. More implementation details of these models are demonstrated in Appendix A.

## 6.2 Main Results

Table 2 presents the pass rate of different models on TRIGO-real (training, validation, and test sets), TRIGO-web (test set only), and TRIGO-gen (training, validation, and test sets). The pass rate indicates the percentage of problems that a model outputs a correct proof within the maximum search steps by interacting with Lean-Gym. All models are trained and evaluated on the corresponding training and test splits, except for the TRIGO-real which the models are trained on TRIGO-real training split and tested on the TRIGO-real test split and TRIGO-web.

In Table 2 we find that: (1) Models with different parameter sizes (GPT-2$_B$, GPT-2$_L$) achieve similar performance on both TRIGO-real and TRIGO-web when trained on the smaller dataset TRIGO-real. On TRIGO-gen that has more training samples, larger model parameter scales lead to better performance; (2) GPT-2$_L$-PACT achieves the best results on each test set, indicating the significant improvement of pre-training on PACT and raising the question of whether we can achieve a similar improvement if fine-tune on TRIGO-gen.

To study the above question, we merge the generated dataset TG-1, TG-2, and TG-3 training split to fine-tune GPT-2$_B$, GPT-2$_L$, and further train GPT-2$_L$-PACT. We denote the resulting models as

GPT-2$_{B-D}$, GPT-2$_{L-D}$, and GPT-2$_L$-PACT-D and evaluate their pass rate. We find that only GPT-2$_L$-PACT-D pre-trained on PACT can obtain significant improvement on TRIGO-gen. However, when we continue to train GPT-2$_L$-PACT-D on TRIGO-real, the performance does not improve significantly, achieving accuracies of only 23.25% and 13.02% on TRIGO-real and TRIGO-web, respectively.

To explore the gap between TRIGO-real and TRIGO-gen, we train the GPT-2$_L$-PACT on TG-E whose samples are generated start with expression in TRIGO-real. The results are denoted as GPT-2$_L$-PACT-E. Compared with GPT-2$_L$-PACT, GPT-2$_L$-PACT-E achieves a 2.33% improvement on the TRIGO-real test set but a 0.22% decrease on TRIGO-web. These results suggest that solely increasing the proof length in the training data using a generation program does not enhance model performance on the TRIGO-web. We posit that this is due to the significant distribution gap between TRIGO-real and TRIGO-web, making it challenging for the data generated based on TRIGO-real to generalize to TRIGO-web.

To investigate the out-of-distribution generalization ability on datasets of varying difficulty, we present the results in Table 3. Table 3 shows the results of only training model on TG-1, TG-2, and TG-3 and testing on other test sets with different distribution. It is shown that all models perform consistently worse than the in-distribution setting. On TG-1, the best GPT-2$_L$-PACT trained on the more complex TG-3 dataset is still 20.1% lower than that trained on TG-1 alone. On TRIGO-real and TRIGO-gen however, model GPT-2$_L$-PACT trained with three generated datasets separately perform worse than the GPT-2$_L$-PACT-E trained on the TG-E. This demonstrates that initiating the automatic theorem generation program with input derived from real-world data effectively bridges the distribution gap between real-world data and generated data.

## 6.3 Model Analysis

In this section, we perform a comprehensive analysis of GPT models on our TRIGO with various settings. We mainly evaluate the PACT pre-trained model as it achieves the best overall performance. **Stepwise Evaluation** We first evaluate the single-step generation performance. We obtain all (goal state, tactic) pairs, and select the pairs whose tactic is not "have" as set w/o *have*. We compare the model's top-1 and top-8 output tactics with the

| Dataset | all | | w/o *have* | |
| --- | --- | --- | --- | --- |
| | EM@1 | EM@8 | EM@1 | EM@8 |
| TRIGO-real | 69.81 | 78.40 | 80.81 | 86.85 |
| TG-1 | **92.22** | **97.31** | **94.33** | **98.56** |
| TG-2 | 89.29 | 93.71 | 93.11 | 96.22 |
| TG-3 | 86.13 | 90.56 | 90.77 | 93.65 |

Table 4: The single step performance of GPT-2$_L$-PACT on different datasets. EM@k represents the exact match scores of the top-k generated tactics.

| Model | TG-1 | TG-2 | TG-3 | TRIGO-real |
| --- | --- | --- | --- | --- |
| GPT-2$_L$-PACT | 90.23 | 85.77 | 81.27 | 45.40 |
| GPT-4 | 16.91 | 7.44 | 4.67 | 0.26 |

Table 5: Exact match scores of single step performance on "have" tactic. We obtain a tactic with highest probability from GPT-2$_L$-PACT, and provide GPT-4 with an 8-shot prompt (randomly sampled from the (state, tactic) pairs in the training set that contain "have" tactic).

ground truth and consider the model is correct if any of the generated tactics is an exact match with the ground truth. The results are shown in Table 4.

GPT models achieve high performance on tactics without the "have" tactics, with EM scores above 86% for the top 8 generated tactics. However, the prediction of "have" tactics poses a significant challenge in overall proof generation, especially in the TRIGO-real dataset where there is an 8.45% gap in EM@8 between "all" tactics and tactics without the "have" tactics. In the TRIGO-gen dataset, the gap in EM@8 between "all" tactics and tactics without the "have" tactics increases as the number of proof steps increases.

To explore the impact of model size on the accuracy of single-step proofs, we evaluate GPT-4. We believe that LLMs such as GPT-4 have already demonstrated their ability to process formal language, particularly in translating informal proofs to formal proofs (Jiang et al., 2023; Wu et al., 2022). Furthermore, LLMs have demonstrated promise in theorem proving using proof assistants such as Lean, as evidenced by Yang et al.'s study (Yang et al., 2023). The study showed that GPT-4 could generate proofs accepted by a zero-shot manner, thus establishing GPT-4 as a robust baseline. These results suggest that GPT-4 may have been trained on Lean examples, as there were publicly accessible proofs on GitHub prior to GPT-4's data cutoff date in September 2021 (OpenAI, 2023).

We also evaluate the single-step performance of GPT-4, focusing specifically on the "have" tactics. Since the "have" tactic only requires generating the

| Goal | GOAL
⊢ cos(433*pi/3)
= 2*cos(pi/6)**2
- 1 **PROOFSTEP** | GOAL h0 : tan(295*pi/6) ≠ 0,
this : tan(-pi/3)
= (-1)/tan(295*pi/6)
⊢ sin(-pi/3)/cos(-pi/3)
= tan(-pi/3) **PROOFSTEP** | GOAL ⊢ sin(4*pi/3)*cos(pi/3)
- sin(pi/3)*cos(4*pi/3)
= 2*sin(pi/2)*cos(pi/2)
**PROOFSTEP** |
|---|---|---|---|
| GPT-4
w/ ICL | **have**: cos(2*pi/3) =
2*cos(pi/6)**2 - 1, | **have**: sin(-pi/3)/cos(-pi/3) =
tan(-pi/3), | **have**: sin(4pi/3)*cos(pi/3)
- sin(pi/3)*cos(4pi/3) =
2*sin(pi/2)*cos(pi/2), |
| GPT-4
w/ Instruction | **have**: cos(433*pi/3) =
cos((433 mod 6)*pi/3) | **have**: tan(pi/3) =
-tan(113*pi/3) | **have**: sin(2*pi) = sin((-2)*pi) |
| GT | **have**: cos(pi/3) =
cos(433*pi/3), | **have**: sin(-pi/3)/cos(-pi/3) =
tan(-pi/3), | **have**: sin(pi) =
sin(4*pi/3)*cos(pi/3) -
sin(pi/3)*cos(4*pi/3), |

Table 6: One-step proofs generated by GPT-4 given in-context learning (ICL) or natural language instruction (Instruction) and a new goal (Goal), compared with the ground truth (GT).

sub-goal equation without additional Lean knowledge, evaluating GPT-4's performance on these tactics effectively reflects its ability in complex number combination reasoning. For each generation, we randomly select 8 (goal state, tactic) pairs from the training set as in-context learning examples for GPT-4. Table 5 presents the exact match scores obtained. The experimental results highlight a significant performance gap between GPT-4 and the fine-tuned smaller GPT-2 models across all settings. Table 6 showcases several one-step proofs generated by GPT-4.

**Search Evaluation** We conduct three experiments to study the effects of tactics decoding and proof search methods.

We first compare beam search and sampling. When generating tactics, we apply beam search with size 16 and expand the proof goal with the top-8 tactics. As for sampling, we randomly select each token based on the model's output probability and sample 8 tactics. Table 7 shows the proof pass rate, and sampling achieves better performance. After inspecting model outputs, we further observe that sampling produces more diverse tactics, exploring various search paths, and is particularly effective in discovering number combinations.

We then explore the effect of increasing sampling temperatures. With a temperature of 1.5, the model generates many illegal characters but outputs more diverse tactics. Reducing the temperature to 1.25 significantly decreases illegal characters, improving the model's pass rate on TRIGO-real, TG-1, and TG-2. These results suggest a future direction of developing decoding methods to generate diverse and valid tactics.

To compare different search algorithms, we lastly implement breadth-first search (BFS) and Monte Carlo tree search (MCTS) in previous work (Silver et al., 2017). Surprisingly, MCTS does not excel on our dataset. In Table 9, MCTS per-

| Decoding Method | Manual Labeling | | TRIGO-gen | | |
| | TRIGO-real | TRIGO-web | TG-1 | TG-2 | TG-3 |
|---|---|---|---|---|---|
| Beam Search | 23.25 | 20.75 | 68.49 | 38.39 | 6.79 |
| Sampling | **32.55** | **25.60** | 77.29 | **44.19** | **18.99** |

Table 7: Pass rate at different decoding methods.

| Temperature | Manual Labeling | | TRIGO-gen | | |
| | TRIGO-real | TRIGO-web | TG-1 | TG-2 | TG-3 |
|---|---|---|---|---|---|
| 1.0 | **32.55** | 25.60 | 77.29 | 44.19 | **18.99** |
| 1.25 | 31.39 | **26.49** | **77.49** | **45.99** | 18.59 |
| 1.5 | 29.06 | 22.07 | 76.49 | 44.89 | 17.39 |

Table 8: Pass rate with different sampling temperatures.

forms significantly worse than BFS on the artificially synthesized dataset TRIGO-gen. We suppose this is due to the lack of a well-developed value function that should be addressed in future work.

**Expert Iteration** Figure 5 demonstrates the model performances by expert iterations. Generated samples in TRIGO-gen usually have multiple proof paths, thus we apply the expert iteration (Polu et al., 2023) to discover diverse and better proof paths. Specifically, we train the GPT-2$_L$-PACT model on TG-$i$, where $i \in [1, 3]$. Then we employ the trained models to prove the training set samples in TG-$i$. If the proof pass, we add the new proof to the original TG-$i$. Eventually, we expand the original set to a new training set TG$^1$-$i$, where 1 indicates the 1st iteration. We retrain the GPT-2$_L$-PACT on TG$^1$-$i$, generate new proofs, add the new proofs to TG$^1$-$i$ and obtain TG$^2$-$i$. We repeat the above process 7 times, obtaining TG$^1$-$i$ to TG$^7$-$i$. We train the model on the seven datasets and evaluate them on the original test set of TG-$i$. As shown in Figure 5, the model's pass rate improves significantly across all three TRIGO-gen datasets, with the largest improvement of 10.9% in TG-2. This highlights the diversity of proof path in training data for enhancing model performance.

**Large Angle Values Evaluation** To evaluate the model's out-of-distribution (OOD) generalization

| Search Method | Manual Labeling | | TRIGO-gen | | |
|---|---|---|---|---|---|
| | TRIGO-real | TRIGO-web | TG-1 | TG-2 | TG-3 |
| MCTS | **32.55** | 24.06 | 51.59 | 11.49 | 5.89 |
| BFS | **32.55** | **25.60** | **77.29** | **44.19** | **18.99** |

Table 9: Pass rate at different search methods.

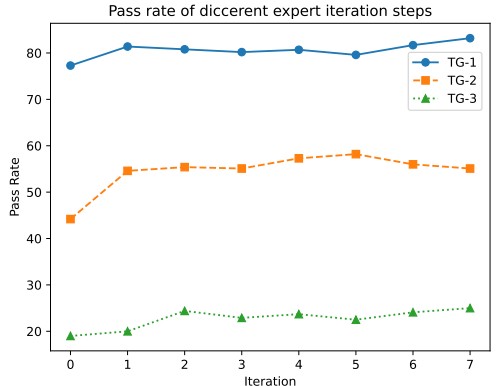

Figure 5: The accuracy of the GPT-$2_L$-PACT model on the test set during expert iteration.

ability on numerical reasoning, we expand the range of angle values $C$ that can be sampled during generation to a more complex set $C_l = \{2\pi, \pi, \frac{\pi}{2}, \frac{\pi}{3}, \frac{\pi}{6}, -\frac{\pi}{6}, -\frac{\pi}{3}, -\frac{\pi}{2}, -\pi, -2\pi, \frac{\pi}{4}, \frac{\pi}{5}, \frac{\pi}{7}, \frac{\pi}{8}, \frac{\pi}{9}, -\frac{\pi}{4}, -\frac{\pi}{5}, -\frac{\pi}{7}, -\frac{\pi}{8}, -\frac{\pi}{9}\}$, and extend the maximum value of $K$ from 100 to 1000. We generate harder test sets for TG-1/2/3 respectively. We find that all models including the strongest baseline GPT-$2_L$-PACT-D model achieve a pass rate of $0$ on these OOD test sets, revealing the limitation of current language models on the numerical reasoning.

## 7 Conclusion

In this paper, we introduce TRIGO, a dataset focusing on trigonometric expression reduction for formal mathematical reasoning with both real-world and generated samples. To the best of our knowledge, TRIGO is the first Lean-based dataset with manually annotated and automatically generated reduction proofs for exploring the formal mathematical ability of current language models.

Our comprehensive experiments reveal that, in comparison to generated data, pre-training on PACT significantly enhances performance on real-world problems. Furthermore, expanding the data scale by utilizing real-world data as the starting point for the theorem generation program can effectively boost the model's performance on the real-world test set. Additionally, we reveal the model's incapability on generalizing numeric operations to larger unseen numbers and find that both the di-

versity of tactics and search paths have significant impacts on the final proof pass rate.

## 8 Ethics Statement

The trigonometric expression reduction dataset TRIGO is obtained from the Internet. After we collect the data, we formalize it in Lean and submit it to Lean to verify the correctness of the proof, without any bias involved.

When annotating TRIGO-real, we utilize not only the publicly available answers from "tiku", but also compose some of the answers ourselves. As for collecting unlabeled data for TRIGO-web, we make efforts to gather solutions from diverse sources, encompassing blogs, documentation, Q&A communities, and even videos.

## 9 Limitations

Our evaluation metric focuses solely on verifying the correctness of the model's proofs. We consider the "**no goals**" output in the interactive environment Lean-Gym as an indication of success. Hence, this indicator serves as our metric for assessing the model's performance. In our future work, we aim to introduce improved evaluation metrics to assess the model's ability to generate a wider range of proof paths.

Due to regional constraints, we cannot access the services offered by OpenAI, such as GPT-4 and GPT-3.5. Therefore, the evaluation of GPT-4 and GPT-3.5 in our paper has been entrusted to researchers from a research institution outside the restricted region of OpenAI, who conducted the assessment for this part.

## 10 Acknowledgements

This work was supported in part by National Key R&D Program of China under Grant No. 2020AAA0109700, NSFC Grant No. 62276002, NSFC Grant No.62006255, Guangdong Outstanding Youth Fund (Grant No. 2021B1515020061), NSFC Mobility Grant Award under Grant No. M-0461, Shenzhen Science and Technology Program (Grant No. RCYX20200714114642083), Shenzhen Science and Technology Program (Grant No. GJHZ20220913142600001), Nansha Key RD Program under Grant No.2022ZD014 and Sun Yatsen University under Grant No. 22lgqb38 and 76160-12220011. We thank MindSpore for the partial support of this work.

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

## A Implementation Details.

We employ identical hyperparameters to train GPT models on both the PACT and TRIGO datasets. The Adam optimizer (Kingma and Ba, 2015) is utilized with a learning rate of $2.5 \times 10^{-4}$ and a cosine schedule, while the batch size is set to 512. During training on TRIGO, we set a maximum epoch limit of 20 and select the epoch that achieves the lowest validation set loss. For PACT training, we conduct an initial pre-training epoch on the mathlib, mix1, and mix2 datasets provided by PACT. During the proof search phase, beam search is applied to generate tactics with a beam size of 16, and we consider the top 8 outputs. Additionally, a maximum budget of 512 search steps is allotted. All experiments are executed on 8 Nvidia Tesla V100 GPUs.

## B Case Study

We provide an example of the model's search on TG-1 in this section. As shown in Figure 8, this example demonstrates a correct proof step, and we can see that the model employs the tactic of "have" to make multiple assumptions. However, often the goal of many hypotheses to prove is trivial, so determining how to make useful assumptions is a crucial factor in the model's proof accuracy. Furthermore, the model's ability to combine the numbers in the "have" tactic also determines whether the model can reach the correct proof path.

As shown in Figure 9, this example demonstrates a proof step where the model fails. The model appears to struggle to output diverse hypotheses to explore more proof paths when forming a search tree but generates a large number of identical "have" tactics. It highlights the importance of using a variety of exploration strategies to improve accuracy.

As shown in Table 10, GPT-4 with the In-context learning approach performs well in both the second and third examples, while methods using natural language instructions fail completely. These examples illustrate that GPT-4 is capable of learning the compositional relationships in numbers through in-context learning in one-step proofs.

## C Experimental Details of Monte Carlo Tree Search

We provide here the details of our implementation of Monte Carlo Tree Search. We follow the formula below to generate each tactic $t^*$:

$$\text{PUCT}(g,t) = Q(g,t) + c \cdot P_\theta(t \mid g) \cdot \frac{\sqrt{\sum N(g,\cdot)}}{1 + C(g,t)}, \quad (2)$$
$$t^* = \arg\max_{t \in \mathcal{A}} \text{PUCT}(g,t),$$

where $Q(g,t)$ denotes the value function of sampling tactic $t$ in proof state $g$, $A$ denotes all the tactics that can be sampled, $P_\theta(t \mid g)$ denotes the prior probability, and $C(g,t)$ denotes the number of visit counts of the sampled tactic $t$ in state $g$. In our experiments, we always set the constant $c$ to 1, and we use the cumulative probability of the model output tactic as the value of $Q(g,t)$.

## D Informal Mathematics Benchmarks

In contrast to formal benchmarks, informal benchmarks lack data annotated in a formal theorem proving language. Constructing formal mathematics is time-consuming and demands a high level of mathematical expertise from contributors. Informal math problem datasets, represented in natural language, are more convenient to construct. Math word problems (Koncel-Kedziorski et al., 2016; Wang et al., 2017; Patel et al., 2021; Cobbe et al., 2021; Xiong et al., 2022, 2023; Yu et al., 2023) target elementary students, querying an unknown variable given a natural language situation description. MATH (Hendrycks et al., 2021) contains 12,500 high school math competition problems with natural language statements and solutions. These datasets, collected from real human problem-solving, better reflect real distribution but lack strict formal verification to ensure correctness. NaturalProofs (Welleck et al., 2021) uses natural language to describe mathematical statements and proofs, while (Saxton et al., 2019) synthetically generates sequence-to-sequence math problems represented as pure strings, covering various topics. However, due to natural language ambiguity, the correctness of the proof process in these works cannot be verified.

## E Neural Theorem Proving

DeepHOL (Bansal et al., 2019b) first applies reinforcement learning to automatic theorem proving without human-written proofs, achieving the best performance on HOList. AStactic (Yang and Deng, 2019) treats tactics as programs and composes abstract syntax trees (ASTs) during tactic generation. LIME (Wu et al., 2021b) introduces inductive bias

| | | | |
|---|---|---|---|
| In-context | **GOAL** ⊢ `-cos(154*pi/3) = cos(131*pi/3)` **PROOFSTEP have:** `cos(pi/3) = -cos(154*pi/3),`
**GOAL** ⊢ `2 * sin pi * cos (pi / 3) * cos pi = sin (5 * pi / 3) / 2 + sin (7 * pi / 3) / 2`
  **PROOFSTEP have:** `2*sin(pi)*cos(pi)*cos(pi/3) = 2*sin(pi)*cos(pi)*cos(pi),`
**GOAL** `h0 : tan (66 * pi) ≠ 0 ⊢ 1 / tan (66 * pi) = tan (187 * pi / 2)`
  **PROOFSTEP have:** `tan(-pi/2) = 1 / tan(66*pi),`
**GOAL** ⊢ `-cos (69 * pi / 2) = -sin (34 * pi)` **PROOFSTEP have:** `sin(pi) = -cos(69*pi/2),`
**GOAL** `h0 : sin (pi / 3) ≠ 0, h1 : 2 * sin (pi / 3) ≠ 0, h2 : sin (2 * pi / 3) / (2 * sin (pi / 3))`
  `≠ 0, h3 : sin (2 * pi / 3) ≠ 0 ⊢ tan (pi / 3) =`
  `2 * sin (pi / 3) ** 2 / sin (2 * pi / 3)`
  **PROOFSTEP have:** `sin(pi/3)/(sin(2*pi/3) / (2*sin(pi/3))) = 2*sin(pi/3)**2/sin(2*pi/3),`
**GOAL** `this : cos (pi / 6) = 2 * cos (pi / 12) ** 2 - 1 ⊢ sin ((-2) * pi / 3)`
  `= sin (-pi / 2) * cos (pi / 6) - sin (pi / 6) * cos (-pi / 2)`
  **PROOFSTEP have:** `sin(-2*pi/3) = sin(-pi/2) * cos(pi/6) - sin(pi/6) * cos(-pi/2),`
**GOAL** ⊢ `cos (-pi) = sin ((-133) * pi / 2)` **PROOFSTEP have:** `- -sin((-133) * pi / 2) = sin(-133*pi/2),`
**GOAL** ⊢ `sin ((-4) * pi) = (-2) * sin ((-2) * pi) * sin (299 * pi / 2)`
  **PROOFSTEP have:** `2*sin((-2)*pi) * -sin(299*pi/2) = -2*sin(-2*pi)*sin(299*pi/2),` | | |
| | **GOAL**
⊢ `cos(433*pi/3)`
  `= 2*cos(pi/6)**2`
  `- 1` **PROOFSTEP** | **GOAL** `h0 : tan(295*pi/6) ≠ 0,`
  `this : tan(-pi/3)`
  `= (-1)/tan(295*pi/6)`
  ⊢ `sin(-pi/3)/cos(-pi/3)`
  `= tan(-pi/3)` **PROOFSTEP** | **GOAL** ⊢ `sin(4*pi/3)*cos(pi/3)`
  `- sin(pi/3)*cos(4*pi/3)`
  `= 2*sin(pi/2)*cos(pi/2)`
  **PROOFSTEP** |
| GPT-4 | **have:** `cos(2*pi/3) =`
`2*cos(pi/6)**2 - 1,` | **have:** `sin(-pi/3)/cos(-pi/3) =`
`tan(-pi/3),` | **have:** `sin(4pi/3)*cos(pi/3)`
`- sin(pi/3)*cos(4pi/3) =`
`2*sin(pi/2)*cos(pi/2),` |
| Instructions | `You are an expert in Lean.`
`Now I will give you a`
`one-step goal-proof pair:`
**GOAL** ⊢ `-cos(154*pi/3) =`
`cos(131*pi/3)` **PROOFSTEP**
**have:** `cos(pi/3) =`
`-cos(154*pi/3), Please`
`provide a tactic that`
`includes the word "have"`
`as shown in the example`
`above:` **GOAL** ⊢ `cos (433 *`
`pi / 3) = 2 * cos (pi / 6)`
`** 2 - 1` **PROOFSTEP** | `You are an expert in Lean.`
`Now I will give you a`
`one-step goal-proof pair:`
**GOAL** ⊢ `-cos(154*pi/3) =`
`cos(131*pi/3)` **PROOFSTEP have:**
`cos(pi/3) = -cos(154*pi/3),`
`Please provide a tactic`
`that includes the word`
`"have" as shown in the`
`example above:` **GOAL** `h0 :`
`tan(295*pi/6)≠0, this :`
`tan(-pi/3)=(-1)/tan(295*pi/6)`
⊢ `sin(-pi/3)/cos(-pi/3) =`
`tan(-pi/3)` **PROOFSTEP** | `You are an expert in Lean. Now`
`I will give you a one-step`
`goal-proof pair:` **GOAL** ⊢
`-cos(154*pi/3) = cos(131*pi/3)`
**PROOFSTEP have:** `cos(pi/3)`
`= -cos(154*pi/3), Please`
`provide a tactic that includes`
`the word "have" as shown`
`in the example above:` **GOAL**
⊢ `sin(4*pi/3)*cos(pi/3) -`
`sin(pi/3)*cos(4*pi/3) =`
`2*sin(pi/2)*cos(pi/2)` **PROOFSTEP** |
| GPT-4 | **PROOFSTEP have:**
`cos(433*pi/3) = cos((433`
`mod 6)*pi/3)` | **PROOFSTEP have:** `tan(pi/3) =`
`-tan(113*pi/3)` | **PROOFSTEP have:** `sin(2*pi) =`
`sin((-2)*pi)` |
| GT | **have:** `cos(pi/3) =`
`cos(433*pi/3),` | **have:** `sin(-pi/3)/cos(-pi/3) =`
`tan(-pi/3),` | **have:** `sin(pi) =`
`sin(4*pi/3)*cos(pi/3) -`
`sin(pi/3)*cos(4*pi/3),` |

Table 10: One-step proofs from 8-shot in-context demonstrations generated by GPT-4. We compare results from in-context demonstrations and natural language instructions.

of reasoning into language models through synthetic tasks. GPT-$f$ (Polu and Sutskever, 2020) is the first work leveraging pre-trained language models for automatic theorem proving, generating tactics, and proposing proof search expansion for efficient proof tree searching. Our experiments primarily reference GPT-$f$'s design. PACT (Han et al., 2021) extends this line of work through co-training generating models with multiple self-supervised auxiliary tasks, providing the strongest baseline model after training on PACT's pre-training corpus. Lample et al. (2022) extends MCTS (Abramson and Korf, 1987) to hypertrees, proposing Hyper-Tree Proof Search for theorem proving. Polu et al. (2023) explores curriculum learning for performance improvement, which we incorporate through expert iterations in our study. Xin et al. (2023) explores the utilization of a growing skill library to augment the theorem proving capabilities of large

language models, allowing them to generate novel skills and enhance success rates in mathematical theorem proving tasks, thereby paving the way for new avenues in the theorem proving community.

## F More Details on Automatic Sample Generation

As illustrated in Algorithm 1, we have devised our automated sample generation program by drawing inspiration from the manual annotation process of problem-solving, as depicted in Figure 7. Lean-Gym implements a strict replacement strategy and does not perform reduction operations like sympy. Specifically, when performing the expression replacement step, since we use sympy to parse the expression tree, it automatically reduces the expression to the new equation $eq_t$, causing misalignment with the equation $eq_{lean}$ in Lean's proof goal. To solve this problem, we use the "have" tac-

| Types of tactics | TRIGO-real | TG-1 | TG-2 | TG-3 |
|---|---|---|---|---|
| all tactics | 10,574 | 209,662 | 349,393 | 498,417 |
| *have* | 2,179 | 32,490 | 53,955 | 76,413 |
| Ratio | 0.206 | 0.155 | 0.154 | 0.153 |

Table 11: Statistics for Ratio of "have" tactic

| TRIGO-real | TG-1 | TG-2 | TG-3 | $d_{\theta_E}$ |
|---|---|---|---|---|
| 37 | 22 | 35 | 49 | 81 |

Table 12: Average number of tactics per dataset.

tic for alignment, for example: "have $e_{lean}=e_p$, try field_simp at *, try repeat left, tryring, conv to_lhs, rw this", where $e_{lean}$ and $e_p$ respectively represent the side of the equation $eq_{lean}$ and $eq_t$ that is to be replaced. These tactics automatically align the proof target in Lean's proof goal to $eq_t$. We present our generation algorithm in Algorithm 1. The algorithm continuously iterates in the forward process, randomly sampling rules and parameters to generate problems at each iteration, and interacts with Lean-Gym to ensure correctness. Once the specified number of replacement rules or the maximum number of iterations is reached, the algorithm stops and collects tactics in reverse to obtain training data.

## G  Dataset Statistics

In this section, we present the average proof length for TRIGO, along with the split of the training, validation, and test sets in TRIGO-real. We show the proportion of different tactics in Figure 6. Additionally, we display the proportion of occurrences containing the "have" tactic in Table 11. Table 12 shows the average lengths of the datasets in TRIGO. Table 13 shows our split on TRIGO-real.

We can observe that TG-2 has the closest average length to the TG-real. From Table 11, the proportion of occurrences containing the "have" tactic in TG-2 is smaller than that in TRIGO-real. Although the average proof length is different in TRIGO-gen, the occurrence of the "have" tactic is roughly the same. Additionally, upon closer examination of the data, we find that the "have" tactics in TG-3 is comparable in complexity to the "have" tactic in some of the data in TRIGO-real. The aforementioned statistical data reflects the disparity between the distribution of generated data and real-world data.

| | TRIGO-real | | TG-1 | TG-2 | TG-3 |
|---|---|---|---|---|---|
| Split | Problem | Tactic | Problem | Tactic | Tactic | Tactic |
| Train | 299 | 7,338 | 9,000 | 115,475 | 189,138 | 274,373 |
| Val | 42 | 1,212 | 1,000 | 13,819 | 22,220 | 31,660 |
| Test | 86 | 2,024 | 1,000 | 14,285 | 22,979 | 32,021 |
| All | 427 | 10,574 | 11,000 | 143,579 | 234,337 | 338,054 |

Table 13: Statistics for TRIGO-real.

## H  General Tactic

In this section, we present several tactics we generated along with their corresponding annotations[3]. Table 14 shows several typical tactics, such as "field_simp", both of which are high-level tactics that can handle many expressions involving field operations such as addition, multiplication, and inverse. We do not consider "tidy" when generating the data because this tactic is easily timed out.

## I  Rule Specifications

In this section, we present all the rules manually defined in Table 15-17. For each rule, we create corresponding tactics to ensure that the proof of the problem can be collected backward after generating the corresponding tactics forward. We demonstrate the mapping of some rules to their corresponding tactics in Table 38. During the complete process of annotating these missing reduction steps, as demonstrated in Figure 7, we illustrate how it is matched with the rule specifications, undergoes a one-step transformation, and results in a new problem state.

## J  Annotator Demographics

Our annotation team consists of four Master's students and three PhD students. The Master's students are responsible for completing the annotation of informal proof steps, while the Ph.D students focuse on formalizing the informal proofs into Lean. The four Master's students holds a Master's degree in Computer Science. They have received rigorous mathematical training and possess in-depth knowledge in the process of reducing trigonometric expressions. The three Ph.D students are in the field of computer science specializing in formal theorem proving. Four Master's students firstly simplify the trigonometric functions, then Ph.D students manually translate these informal proofs to Lean and submit them to the Lean theorem prover for correctness checking. The combined process of data annotation and mathematical formalization took one month to complete. The personnel involved in

---

[3] https://leanprover.github.io/documentation/

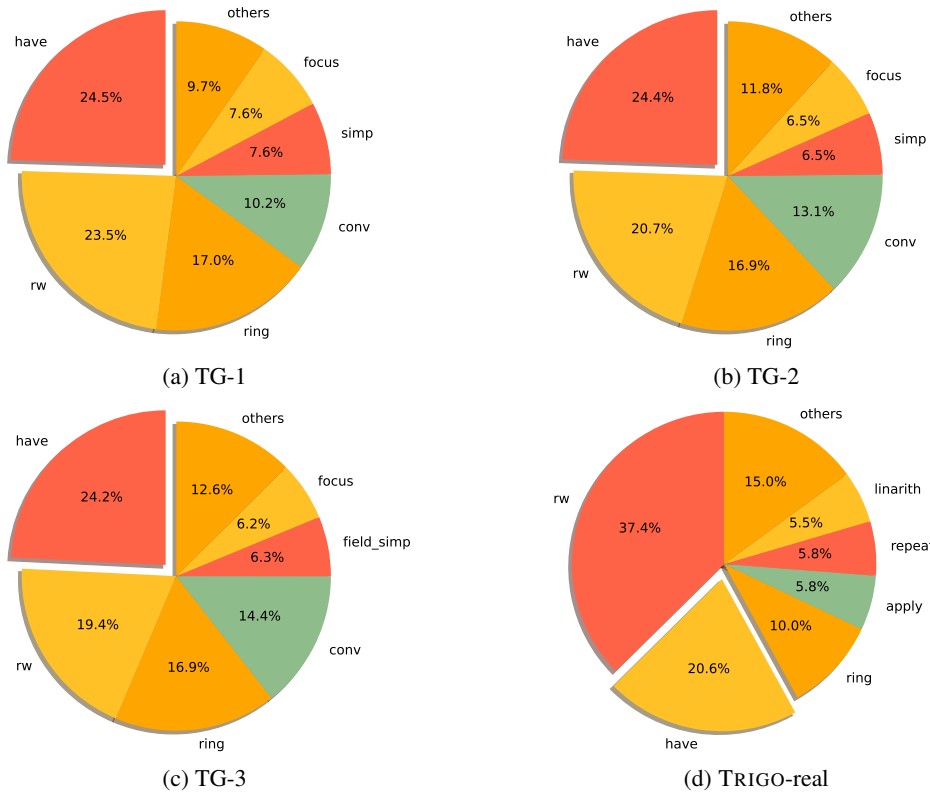

(a) TG-1                    (b) TG-2

(c) TG-3                    (d) TRIGO-real

Figure 6: Tactic distribution of TG-1, TG-2, TG-3, TRIGO-real

the formalization work are also co-authors of our work.

## K  Data Example

We show examples from our dataset in this section. Table tables 18 to 25 shows typical examples in TRIGO. This data can be compiled correctly within Lean-Gym, and after compilation, we interact with Lean-Gym to obtain the final training data.

## L  Large Language Model test examples

In this section, we demonstrate examples of in-context learning and zero-shot methods on large language models. In contrast to the single-step proof context learning in Table 6, the examples of context learning presented in this section directly provide the entire proof. We find that it is challenging for the model to provide correct proof. Tables 30, 32, 26, and 28 show examples of our in-context learning approach tested on our dataset using GPT-3.5 and GPT-4 models. Tables 31, 33, 27, and 29 show the outputs of these models. We find that LLMs have difficulty learning to use the correct "have" tactic, which suggests that LLMs may not be able to manipulate numbers and lack generalization abilities such as the commonly used

techniques of grouping and factoring in trigonometric reduction.

We conduct numerous zero-shot tests on our dataset using GPT-3.5 and GPT-4, employing the prompt "Please help me prove the following lemma using Lean: lemma Trigo_0 : [PROBLEM] :=". However, we find that GPT-4 tends to output a lot of meaningless tactics, such as the case shown in Table 35. Moreover, GPT-4 often generates tactics that do not exist in the dependencies, as seen in 35. The example in Table 35 reveals that GPT-4 is prone to making trivial assumptions. By comparing the output of GPT-3.5 and GPT-4 in tables 34 to 37, we observe that GPT-4 is more inclined to generate "have" tactics, which is closer to the proof pattern of our dataset.

The experiments above all indicate the limitations of LLMs on our dataset. We leave to future work on how to enable large language models to acquire complex number combinations ability and reduce illusions.

| Tactic | Function |
| --- | --- |
| field_simp | The goal of field_simp is to reduce an expression in a field to an expression of the form $n \div d$ where neither $n$ nor $d$ contains any division symbol. |
| simp | In Lean, simp is a tactic that stands for "simplification." It is used to simplify expressions and goals by applying a set of predefined rewrite rules and simplification procedures. The purpose of simp is to automatically transform complex or convoluted expressions into simpler forms, making them easier to work with and reason about. |
| ring_exp | A tactic for solving equations in commutative (semi)rings, where the exponents can also contain variables. |
| ring | Evaluate expressions in the language of commutative (semi)rings. |
| assumption | The assumption tactic looks through the assumptions in context of the current goal, and if there is one matching the conclusion, it applies it. |
| repeat assumption | The repeat assumption tactic looks through the assumptions in context of all goals, and if the assumption of the context of the current goal can match the target, then it is applied. |
| left | The tactic tries to solve the left disjunct immediately by assumption; if that fails, it tries to focus on the right disjunct; and if that doesn't work, it invokes the assumption tactic. |
| refl | In the proof language Lean, refl is an abbreviation for "reflexivity." It is used as a tactic to automatically prove goals of the form a = a, where a is any term or expression. Essentially, it asserts that any term is equal to itself, which is a fundamental property of equality. |
| have | In Lean, "have" is a keyword used in proof scripts to introduce a new intermediate goal or hypothesis. It allows the user to assert a proposition and then prove it separately before continuing with the rest of the proof. |
| conv | In Lean, conv is a tactic that allows users to perform step-by-step rewriting and manipulation of expressions within a proof. It stands for "conversion" and provides a flexible way to apply various rewrite rules, simplify expressions, and rearrange terms. |
| to_lhs | The to_lhs modifier is typically used within a tactic block, such as conv or rewrite, to specify the side of the equation or expression that should be modified. When to_lhs is used, the tactic will focus on the LHS of the equation or expression and perform the specified operations on that side. |
| rw | In Lean, rw is a tactic that stands for "rewrite". It is used to apply a specific rewrite rule to an expression or goal within a proof. The rw tactic is commonly used to replace occurrences of a specified term or pattern with a different term or pattern according to a given equality. |
| apply | In Lean, the apply tactic is used to apply a theorem or a hypothesis as a rule to prove a goal or to generate new subgoals. It allows the user to use an existing proposition to infer or establish other propositions. |
| congr_arg | In Lean, congr_arg is a function that allows users to apply congruence to a function applied to an argument. It is used to prove equalities by reasoning about the effects of a function on its arguments. |

Table 14: Examples of general tactic.

| Identity Rule Name | Example |
| --- | --- |
| sin_zero | $\sin(0) = 0$ |
| sin_pi | $\sin(\pi) = 0$ |
| sin_two_pi_div_three | $\sin\left(\frac{2\pi}{3}\right) = \frac{\sqrt{3}}{2}$ |
| sin_three_pi_div_four | $\sin\left(\frac{3\pi}{4}\right) = \frac{\sqrt{2}}{2}$ |
| sin_five_pi_div_six | $\sin\left(\frac{5\pi}{6}\right) = \frac{1}{2}$ |
| sin_pi_div_twelve | $\sin\left(\frac{\pi}{12}\right) = -\frac{\sqrt{2}}{4} + \frac{\sqrt{6}}{4}$ |
| sin_pi_div_two | $\sin\left(\frac{\pi}{2}\right) = 1$ |
| sin_pi_div_three | $\sin\left(\frac{\pi}{3}\right) = \frac{\sqrt{3}}{2}$ |
| sin_pi_div_four | $\sin\left(\frac{\pi}{4}\right) = \frac{\sqrt{2}}{2}$ |
| sin_pi_div_six | $\sin\left(\frac{\pi}{6}\right) = \frac{1}{2}$ |
| cos_zero | $\cos(0) = 1$ |
| cos_two_pi_div_three | $\cos\left(\frac{2\pi}{3}\right) = -\frac{1}{2}$ |
| cos_three_pi_div_four | $\cos\left(\frac{3\pi}{4}\right) = -\frac{\sqrt{2}}{2}$ |
| cos_five_pi_div_six | $\cos\left(\frac{5\pi}{6}\right) = -\frac{\sqrt{3}}{2}$ |
| cos_pi | $\cos(\pi) = -1$ |
| cos_pi_div_twelve | $\cos\left(\frac{\pi}{12}\right) = \frac{\sqrt{2}}{4} + \frac{\sqrt{6}}{4}$ |
| cos_pi_div_two | $\cos\left(\frac{\pi}{2}\right) = 0$ |
| cos_pi_div_three | $\cos\left(\frac{\pi}{3}\right) = \frac{1}{2}$ |
| cos_pi_div_four | $\cos\left(\frac{\pi}{4}\right) = \frac{\sqrt{2}}{2}$ |
| cos_pi_div_six | $\cos\left(\frac{\pi}{6}\right) = \frac{\sqrt{3}}{2}$ |
| tan_zero | $\tan(0) = 0$ |
| tan_pi | $\tan(\pi) = 0$ |
| tan_two_pi_div_three | $\tan\left(\frac{2\pi}{3}\right) = -\sqrt{3}$ |
| tan_three_pi_div_four | $\tan\left(\frac{3\pi}{4}\right) = -1$ |
| tan_pi_div_twelve | $\tan\left(\frac{\pi}{12}\right) = 2 - \sqrt{3}$ |
| tan_pi_div_three | $\tan\left(\frac{\pi}{3}\right) = \sqrt{3}$ |
| tan_pi_div_four | $\tan\left(\frac{\pi}{4}\right) = 1$ |
| tan_pi_div_six | $\tan\left(\frac{\pi}{6}\right) = \frac{\sqrt{3}}{3}$ |

Table 15: Examples of identities in our pre-defined rule bank. Notations (1/3).

| Identity Rule Name | Example |
| --- | --- |
| sin_neg | $\sin(X) = -\sin(2 * \pi * K - X)$ |
| cos_neg | $\cos(X) = \cos(2 * \pi * K - X)$ |
| tan_neg | $\tan(X) = -\tan(\pi * K - X)$ |
| sin_add_int_mul_two_pi | $\sin(X) = \sin(2 * \pi * K + X)$ |
| cos_add_int_mul_two_pi | $\cos(X) = \cos(2 * \pi * K + X)$ |
| sin_add_int_mul_two_pi_add_pi | $\sin(X) = -\sin(X + \pi * (2 * K + 1))$ |
| cos_add_int_mul_two_pi_add_pi | $\cos(X) = -\cos(X + \pi * (2 * K + 1))$ |
| tan_add_int_mul_pi | $\tan(X) = \tan(\pi * K + X)$ |
| sin_neg_add_int_mul_two_pi_add_pi | $\sin(X) = \sin(-X + \pi * (2 * K + 1))$ |
| cos_neg_add_int_mul_two_pi_add_pi | $\cos(X) = -\cos(-X + \pi * (2 * K + 1))$ |
| sin_add_pi_div_two | $\sin(X) = -\cos\left(2 * \pi * K + X + \frac{\pi}{2}\right)$ |
| sin_add_pi_div_two_add_pi | $\sin(X) = \cos\left(X + \pi * (2 * K + 1) + \frac{\pi}{2}\right)$ |
| sin_neg_add_pi_div_two_add_pi | $\sin(X) = -\cos\left(-X + \pi * (2 * K + 1) + \frac{\pi}{2}\right)$ |
| cos_add_pi_div_two | $\cos(X) = \sin\left(2 * \pi * K + X + \frac{\pi}{2}\right)$ |
| cos_add_pi_div_two_add_pi | $\cos(X) = -\sin\left(X + \pi * (2 * K + 1) + \frac{\pi}{2}\right)$ |
| cos_neg_add_pi_div_two_add_pi | $\cos(X) = -\sin\left(-X + \pi * (2 * K + 1) + \frac{\pi}{2}\right)$ |
| tan_add_pi_div_two | $\tan(X) = -\frac{1}{\tan\left(\pi * K + X + \frac{\pi}{2}\right)}$ |
| sin_neg_add_pi_div_two | $\sin(X) = \cos\left(2 * \pi * K - X + \frac{\pi}{2}\right)$ |
| cos_neg_add_pi_div_two | $\cos(X) = \sin\left(2 * \pi * K - X + \frac{\pi}{2}\right)$ |
| tan_neg_add_pi_div_two | $\tan(X) = \frac{1}{\tan\left(\pi * K - X + \frac{\pi}{2}\right)}$ |
| sin_two_mul | $\sin(2 * X) = 2 * \sin(X) * \cos(X)$ |
| sin_three_mul | $\sin(3 * X) = -4 * \sin^3(X) + 3 * \sin(X)$ |
| cos_two_mul_1 | $\cos(2 * X) = 2 * \cos^2(X) - 1$ |
| cos_two_mul_2 | $\cos(2 * X) = -\sin^2(X) + \cos^2(X)$ |
| cos_two_mul_3 | $\cos(2 * X) = 1 - 2 * \sin^2(X)$ |
| cos_three_mul | $\cos(3 * X) = 4 * \cos^3(X) - 3 * \cos(X)$ |
| tan_two_mul | $\tan(2 * X) = \frac{2 * \tan(X)}{1 - \tan^2(X)}$ |
| sin_sq_cos_two_mul | $\sin(X)^2 = \frac{1 - \cos(2 * X)}{2}$ |
| cos_sq_cos_two_mul | $\cos(X)^2 = \frac{1 + \cos(2 * X)}{2}$ |
| cos_eq_sin_two_mul | $\cos(X) = \frac{\sin(2 * X)}{2 * \sin(X)}$ |
| sin_eq_sin_two_mul | $\sin(X) = \frac{\sin(2 * X)}{2 * \cos(X)}$ |

Table 16: Examples of identities in our pre-defined rule bank. Notations (2/3).

| Identity Rule Name | Example |
|---|---|
| sin_add_sin | $\sin(X) + \sin(Y) = 2 * \sin\left(\frac{X+Y}{2}\right) * \cos\left(\frac{X-Y}{2}\right)$ |
| sin_sub_sin | $\sin(X) - \sin(Y) = 2 * \sin\left(\frac{X-Y}{2}\right) * \cos\left(\frac{X+Y}{2}\right)$ |
| cos_add_cos | $\cos(X) + \cos(Y) = 2 * \cos\left(\frac{X-Y}{2}\right) * \cos\left(\frac{X+Y}{2}\right)$ |
| cos_sub_cos | $\cos(X) - \cos(Y) = -2 * \sin\left(\frac{X-Y}{2}\right) * \sin\left(\frac{X+Y}{2}\right)$ |
| tan_add_tan | $\tan(X) + \tan(Y) = (1 - \tan(X) * \tan(Y)) * \tan(X + Y)$ |
| tan_sub_tan | $\tan(X) - \tan(Y) = (1 + \tan(X) * \tan(Y)) * \tan(X - Y)$ |
| tan_sub_tan_2 | $\tan(X) - \tan(Y) = \frac{\sin(X-Y)}{\cos(X)*\cos(Y)}$ |
| tan_div_two_1 | $\tan(X/2) = \frac{1-\cos(X)}{\sin(X)}$ |
| tan_div_two_2 | $\tan(X/2) = \frac{\sin(X)}{1+\cos(X)}$ |
| sin_mul_sin | $\sin(X) * \sin(Y) = \frac{\cos(X-Y)-\cos(X+Y)}{2}$ |
| sin_mul_cos | $\sin(X) * \cos(Y) = \frac{\sin(X+Y)+\sin(X-Y)}{2}$ |
| cos_mul_sin | $\sin(Y) * \cos(X) = \frac{\sin(X+Y)-\sin(X-Y)}{2}$ |
| cos_mul_cos | $\cos(X) * \cos(Y) = \frac{\cos(X-Y)+\cos(X+Y)}{2}$ |
| tan_mul_tan | $\tan(X) * \tan(Y) = \frac{\tan(X)-\tan(Y)}{\tan(X-Y)} - 1$ |
| tan_mul_tan_2 | $\tan(X) * \tan(Y) = -\frac{\tan(X)+\tan(Y)}{\tan(X+Y)} + 1$ |
| sin_add | $\sin(X + Y) = \sin(X) * \cos(Y) + \sin(Y) * \cos(X)$ |
| sin_sub | $\sin(X - Y) = \sin(X) * \cos(Y) - \sin(Y) * \cos(X)$ |
| cos_add | $\cos(X + Y) = -\sin(X) * \sin(Y) + \cos(X) * \cos(Y)$ |
| cos_sub | $\cos(X - Y) = \sin(X) * \sin(Y) + \cos(X) * \cos(Y)$ |
| tan_add | $\tan(X + Y) = \frac{\tan(X)+\tan(Y)}{1-\tan(X)*\tan(Y)}$ |
| tan_sub | $\tan(X - Y) = \frac{\tan(X)-\tan(Y)}{1+\tan(X)*\tan(Y)}$ |
| tan_eq_sin_div_cos | $\tan(X) = \frac{\sin(X)}{\cos(X)}$ |
| sin_div_cos_eq_tan | $\frac{\sin(X)}{\cos(X)} = \tan(X)$ |
| sin_sq_add_cos_sq | $\sin^2(X) + \cos^2(X) = 1$ |
| sin_sq | $\sin^2(X) = 1 - \cos^2(X)$ |
| cos_sq | $\cos^2(X) = 1 - \sin^2(X)$ |

Table 17: Examples of identities in our pre-defined rule bank. Notations (3/3).

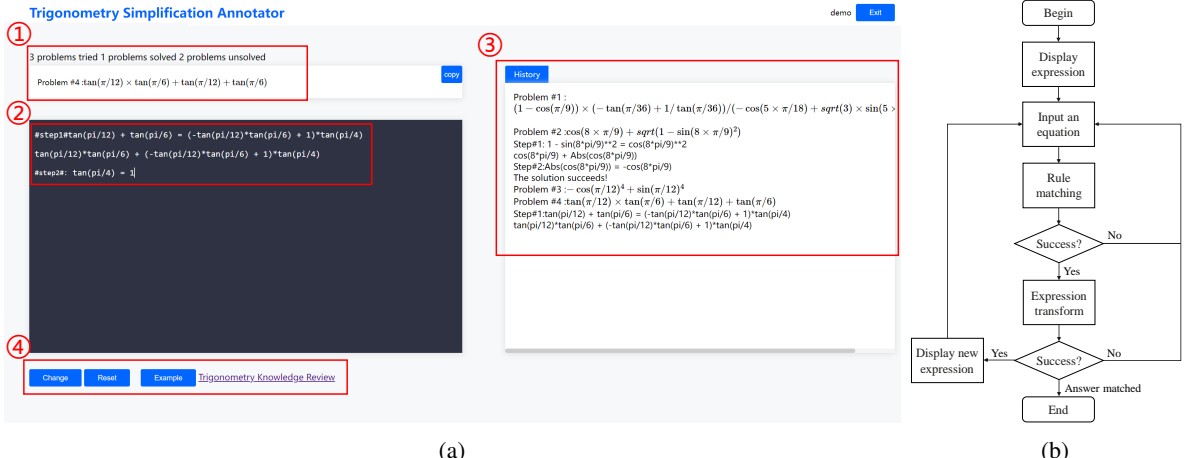

Figure 7: The interactive annotation system for trigonometry reduction. (a) The interface of our system. Region ① shows the problem to be annotated. Region ② is the main interaction area where annotators input an equation for the current step. The system then matches it with the rule bank, performs a one-step transformation, and outputs a new problem state. Region ③ shows the annotation history and region ④ includes interactive buttons for annotators to change or reset the problem, check examples, and trigonometry knowledge to help their annotation. (b) The workflow of our annotation system.

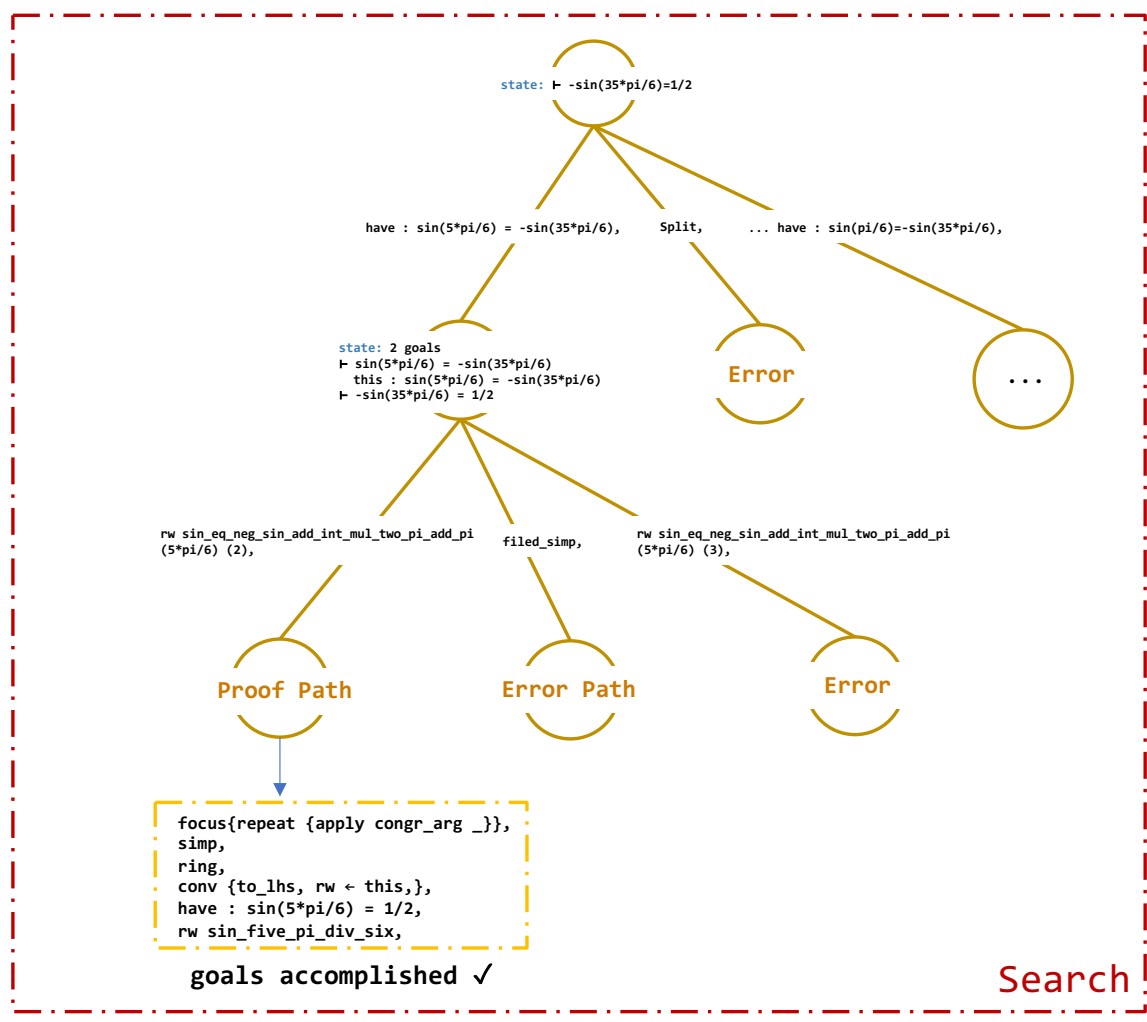

Figure 8: Case 1. The light yellow dotted line indicates the remaining proof steps. Due to page constraints, we do not draw the entire proof tree.

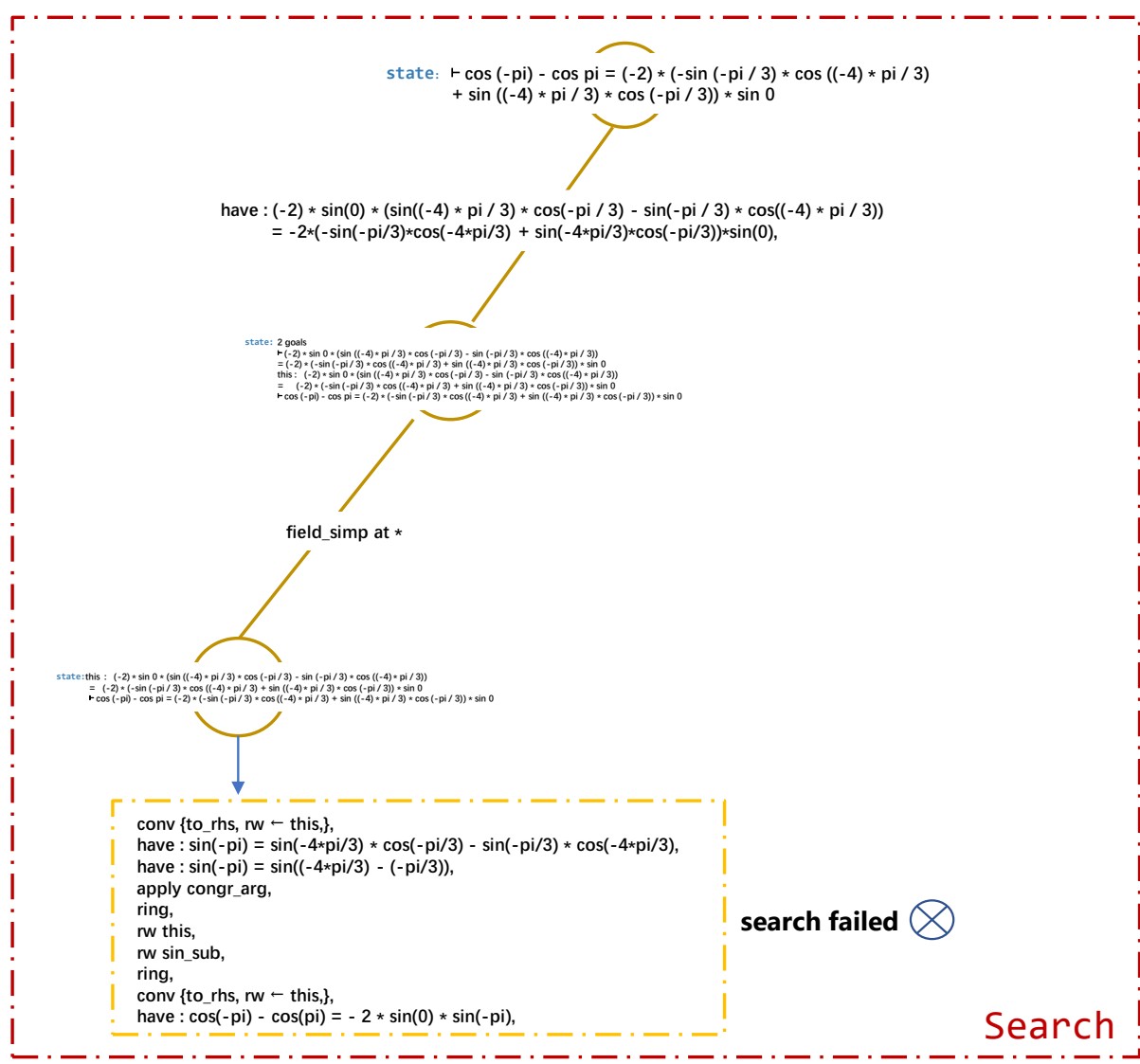

Figure 9: Case 2. The light yellow dotted line indicates the remaining proof steps. Due to page constraints, we do not draw the entire proof tree.

---
**Algorithm 1** Theorem Generator
---
1: **function** GENERATE_THEOREM(len of step $\mathcal{L}$, rule list $\mathcal{R}$)
2:     Randomly select a rule from $\mathcal{R}$:$r_0 \sim Uniform(\mathcal{R})$.
3:     Initialize the parameters $X$, $Y$, $K$ in $r_0$ and get its initialization state in lean and expression: $P_{state}, eq_0 \leftarrow$ INITIALIZE($r_0$).
4:     Init $Tactic_{prove}$, $P_{count}$: $\emptyset, 0 \leftarrow$ INIT()
5:     **for** $t \leftarrow 1$ to $200$ **do**
6:         Randomly select a rule from $\mathcal{R}$:$R_t \sim Uniform(\mathcal{R})$.
7:         Match the formula $R_l$ on the left side of the $R_t$ equation with the formulas $e_l$ and $e_r$ on the left and right sides of $eq_{t-1}$, and return the matching parameters. Since there are multiple matches, a matching result can be randomly selected: $Para_l \leftarrow$ RULE_MATCHING($R_l, e_l$), $Para_r \leftarrow$ RULE_MATCHING($R_l, e_r$).
8:         Substitute the parameters into $R_t$.
9:         **if** $Para_l$ is not NULL **then**
10:             $R_l \leftarrow$ PARAMETER_REPLACEMENT($R_l, Para_l$),
11:             $R_r \leftarrow$ PARAMETER_REPLACEMENT($R_r, Para_l$),
12:         **else if** $Para_r$ is not NULL **then**
13:             $R_l \leftarrow$ PARAMETER_REPLACEMENT($R_l, Para_r$),
14:             $R_r \leftarrow$ PARAMETER_REPLACEMENT($R_r, Para_r$),
15:         **end if**
16:         Perform equation replacement operation: $eq_t \leftarrow$ EQUATION_REPLACEMENT($eq_{t-1}, R_l, R_r$).
17:         Get the tactics corresponding to the rule: $Tactics_{R_t} \leftarrow R_t$.GET_TACTICS($eq_{t-1}, R_l, R_r$).
18:         Obtain the tactic of adjusting the cross terms in $eq_{t-1}$:
19:         **if** $Para_l$ is not NULL **then**
20:             $Tactics_{term} \leftarrow$ GET_CROSS_TERM($e_l, R_p$), if $Para_l$ is not NULL.
21:         **else if** $Para_r$ is not NULL **then**
22:             $Tactics_{term} \leftarrow$ GET_CROSS_TERM($e_r, R_p$), elif $Para_r$ is not NULL.
23:         **end if**
24:         Apply $Tactics_{R_t}$ and $Tactics_{terms}$ to LEAN-GYM to obtain the GOAL$_{lean}$ in lean: GOAL$_{lean}$, ERROR $\leftarrow$ APPLY_TACTICS($Tactics_{term}, Tactics_{R_t}$).
25:         **if** ERROR is not NULL **then**
26:             continue
27:         **end if**
28:         Align the proof goal equation $eq_{lean}$ in GOAL$_{lean}$ with $eq_t$: $Tactics_{align} \leftarrow$ GET_CROSS_TERM($eq_{lean}, eq_t$).
29:         Apply $Tactics_{align}$ to lean: GOAL$_t$, ERROR $\leftarrow$ APPLY_TACTICS($Tactics_{align}$).
30:         **if** ERROR is not NULL **then**
31:             continue
32:         **end if**
33:         $Tactic_{prove} \leftarrow Tactic_{prove} \cup Tactics_{align} \cup Tactics_{term} \cup Tactics_{R_t}$.
34:         $P_{count} \leftarrow P_{count} + 1$.
35:         **if** $P_{count} = \mathcal{L}$ **then**
36:             break
37:         **end if**
38:     **end for**
39:     **return** $Tactic_{prove}$
40: **end function**
---

```
lemma Trigo_0_17_FTGL : sin(-19*pi/6)=1/2:=
begin
  have : sin(-19*pi/6)  =  sin(-7*pi/6),
  {
      rw sin_eq_sin_add_int_mul_two_pi (-19*pi/6) (1),
      repeat {apply congr_arg _},
      simp,
      linarith,
  },
  rw this,
  have : sin(-7*pi/6)  =  -sin(7*pi/6),
  {
      rw sin_eq_neg_sin_neg_add_int_mul_two_pi (-7*pi/6) (0),
      repeat {apply congr_arg _},
      simp,
      linarith,
  },
  rw this,
  have : sin(7*pi/6)  =  -sin(pi/6),
  {
      rw sin_eq_neg_sin_add_int_mul_two_pi_add_pi (7*pi/6) (-1),
      repeat {apply congr_arg _},
      simp,
      linarith,
  },
  rw this,
  rw sin_pi_div_six,
  norm_num,
end
```

Table 18: Example 1 of TRIGO-real

```
lemma Trigo_5_36_KOPV
  (h0 : sin(pi/18) ≥ 0)
  (h1 : -cos(pi/18) + sin(pi/18) ≤ 0)
  (h2 : -sin(pi/18) + cos(pi/18) ≠ 0) :
  sqrt(1 - sin(pi/9))/(-sqrt(1 - cos(17*pi/18)**2) + cos(35*pi/18))=1:=
begin
  rw ← sin_sq,
  have : sin(pi/9)  =  2*sin(pi/18)*cos(pi/18),
  {
      have : sin (pi/9) = sin(2*(pi/18)),
      {
          apply congr_arg,
          ring,
      },
      rw this,
      rw sin_two_mul,
  },
  rw this,
  have : 1 - 2*sin(π/18)*cos(π/18) = sin(pi/18)**2 + cos(pi/18)**2
  - 2*sin(pi/18)*cos(pi/18),
  {
      rw sin_sq_add_cos_sq,
  },
  rw this,
  have : sin(pi/18)**2 + cos(pi/18)**2 - 2*sin(pi/18)*cos(pi/18)  =
  (-cos(pi/18) + sin(pi/18))**2,
  {
      ring_exp,
  },
  rw this,
  have : sin(17*pi/18)  =  sin(pi/18),
  {
      rw sin_eq_sin_neg_add_int_mul_two_pi_add_pi (17*pi/18) (0),
      repeat {apply congr_arg _},
      simp,
      linarith,
  },
  rw this,
  have : cos(35*pi/18)  =  cos(pi/18),
  {
      rw cos_eq_cos_neg_add_int_mul_two_pi (35*pi/18) (1),
      repeat {apply congr_arg _},
      simp,
      linarith,
  },
  rw this,
  repeat {rw sqrt_sq_eq_abs},
  rw abs_eq_self.mpr h0,
  rw abs_eq_neg_self.mpr h1,
  norm_num,
  field_simp,
end
```

Table 19: Example 2 of TRIGO-real

```
lemma Trigo_0 : sin(107*pi)=0:=
begin
have : cos(pi/2) = sin(107*pi),
{
    rw cos_eq_sin_add_pi_div_two_add_int_mul_two_pi (pi/2) (53),
    focus{repeat {apply congr_arg _}},
    simp,
    ring,
},
conv {to_lhs, rw ← this,},
have : cos(pi/2) = 0,
{
    rw cos_pi_div_two,
},
rw this,
end
```

Table 20: Example 1 of TG-1

```
lemma Trigo_3 : -sin(-pi/4)**2 + cos(-pi/4)**2=- sin(51*pi):=
begin
have : cos(-pi/2) = -sin(-pi/4) ** 2 + cos(-pi/4) ** 2,
{
    have : cos(-pi/2) = cos(2*(-pi/4)),
    {
        apply congr_arg,
        ring,
    },
    rw this,
    rw cos_two_mul',
    ring,
},
conv {to_lhs, rw ← this,},
have : cos(-pi/2) = - sin(51*pi),
{
    rw cos_eq_neg_sin_add_pi_div_two_add_int_mul_two_pi_add_pi
    (-pi/2) (25),
    focus{repeat {apply congr_arg _}},
    simp,
    ring,
},
rw this,
end
```

Table 21: Example 2 of TG-1

```
lemma Trigo_1 (h0:sin(1187*pi/12)≠ 0) (h1:(2*sin(1187*pi/12))≠ 0):
-sin(1187*pi/6)/(2*sin(1187*pi/12))=sqrt( 2 ) / 4 + sqrt( 6 ) / 4:=
begin
have : -(sin(1187*pi/6) / (2*sin(1187*pi/12))) = -sin(1187*pi/6)
/(2*sin(1187*pi/12)),
{
    field_simp at *,
},
conv {to_lhs, rw ← this,},
have : cos(1187*pi/12) = sin(1187*pi/6) / (2*sin(1187*pi/12)),
{
    have : sin(1187*pi/6) = sin(2*(1187*pi/12)),
    {
        apply congr_arg,
        ring,
    },
    rw this,
    rw sin_two_mul,
    field_simp at *,
    ring,
},
conv {to_lhs, rw ← this,},
have : cos(pi/12) = -cos(1187*pi/12),
{
    rw cos_eq_neg_cos_neg_add_int_mul_two_pi_add_pi (pi/12) (49),
    focus{repeat {apply congr_arg _}},
    simp,
    ring,
},
conv {to_lhs, rw ← this,},
have : cos(pi/12) = sqrt(2)/4 + sqrt(6)/4,
{
    rw cos_pi_div_twelve,
},
rw this,
end
```

Table 22: Example 1 of TG-2

```
lemma Trigo_2 (h0 : cos((4*pi/3)/2)≠ 0) (h1 : (cos(4*pi/3)+1)≠ 0)
(h2 : (1+cos(4*pi/3))≠ 0) : cos(-5*pi/6)/(cos(4*pi/3)+1)=-sqrt(3):=
begin
have : cos((-5)*pi/6)/(cos(4*pi/3)+1) = cos(-5*pi/6)/(cos(4*pi/3)+1),
{
    field_simp at *,
},
have : sin(4*pi/3) = cos(-5*pi/6),
{
    rw sin_eq_cos_neg_add_pi_div_two_add_int_mul_two_pi (4*pi/3) (0),
    focus{repeat {apply congr_arg _}},
    simp,
    ring,
},
conv {to_lhs, rw ← this,},
have : sin(4*pi/3)/(1+cos(4*pi/3)) = sin(4*pi/3)/(cos(4*pi/3) + 1),
{
    field_simp at *,
    repeat {left},
    ring,
},
conv {to_lhs, rw ← this,},
have : tan(2*pi/3) = sin(4*pi/3) / ( 1 + cos(4*pi/3) ),
{
    have : tan(2*pi/3) = tan((4*pi/3)/2),
    {
        apply congr_arg,
        ring,
    },
    rw this,
    rw tan_div_two',
    repeat {assumption},
},
conv {to_lhs, rw ← this,},
have : tan(2*pi/3) = - sqrt( 3 ),
{
    rw tan_two_pi_div_three,
},
rw this,
end
```

Table 23: Example 2 of TG-2

```
lemma Trigo_2 (h0:cos(-pi/2)≠ 0) (h1:(2*cos(-pi/2))≠ 0):-cos(0)*cos(229*pi/2)
+ sin(0) * sin(-pi)/(2*cos(-pi/2))=0:=
begin
have : -cos(0) * cos(229*pi/2) + sin(0) * (sin(-pi) / (2*cos(-pi/2))) = -cos(0)
*cos(229*pi/2) + sin(0)*sin(-pi)/(2*cos(-pi/2)),
{
    field_simp at *,
},
conv {to_lhs, rw ← this,},
have : sin(-pi/2) = sin(-pi) / ( 2 * cos(-pi/2) ),
{
    have : sin(-pi) = sin(2*(-pi/2)),
    {
        apply congr_arg,
        ring,
    },
    rw this,
    rw sin_two_mul,
    field_simp at *,
    ring,
},
conv {to_lhs, rw ← this,},
have : cos(0) * -cos(229*pi/2) + sin(0) * sin(-pi/2) = -cos(0)*cos(229*pi/2)
+ sin(0)*sin(-pi/2),
{
    field_simp at *,
},
conv {to_lhs, rw ← this,},
have : cos(-pi/2) = -cos(229*pi/2),
{
    rw cos_eq_neg_cos_add_int_mul_two_pi_add_pi (-pi/2) (57),
    focus{repeat {apply congr_arg _}},
    simp,
    ring,
},
conv {to_lhs, rw ← this,},
have : sin(0)*sin(-pi/2) + cos(0)*cos(-pi/2) = cos(0)*cos(-pi/2) + sin(0)*sin(-pi/2),
{
    field_simp at *,
},
conv {to_lhs, rw ← this,},
have : cos(pi/2) = sin(0) * sin(-pi/2) + cos(0) * cos(-pi/2),
{
    have : cos(pi/2) = cos((0) - (-pi/2)),
    {
        apply congr_arg,
        ring,
    },
    rw this,
    rw cos_sub,
    ring,
},
conv {to_lhs, rw ← this,},
have : cos(pi/2) = 0,
{
    rw cos_pi_div_two,
},
rw this,
end
```

Table 24: Example 1 of TG-3

```
lemma Trigo_0_7_HOEW_extend : -2*sin(709*pi/12)*cos(709*pi/12)=-1/2:=
begin
have : -(2*sin(709*pi/12) * cos(709*pi/12)) =
-2*sin(709*pi/12)*cos(709*pi/12),
{
    field_simp at *,
},
conv {to_lhs, rw ← this,},
have : sin(709*pi/6) = 2 * sin(709*pi/12) * cos(709*pi/12),
{
    have : sin(709*pi/6) = sin(2*(709*pi/12)),
    {
        apply congr_arg,
        ring,
    },
    rw this,
    rw sin_two_mul,
},
conv {to_lhs, rw ← this,},
have : sin(163*pi/6) = -sin(709*pi/6),
{
    rw sin_eq_neg_sin_add_int_mul_two_pi_add_pi (163*pi/6) (45),
    focus{repeat {apply congr_arg _}},
    simp,
    ring,
},
conv {to_lhs, rw ← this,},
have : sin(23*pi/6) = sin(163*pi/6),
{
    rw sin_eq_sin_neg_add_int_mul_two_pi_add_pi (23*pi/6) (15),
    focus{repeat {apply congr_arg _}},
    simp,
    ring,
},
conv {to_lhs, rw ← this,},
have : sin(23*pi/6) = -sin(pi/6),
{
    rw sin_eq_neg_sin_neg_add_int_mul_two_pi (23*pi/6) (2),
    repeat {apply congr_arg _},
    simp,
    linarith,
},
rw this,
rw sin_pi_div_six,
norm_num,
end
```

Table 25: Example 1 of TG-E

```
lemma Trigo_169 : sin(69*pi/2)=- sin(-pi) ** 2 + cos(-pi) ** 2:=
begin
have : cos(-2*pi) = sin(69*pi/2),
{
    rw cos_eq_sin_add_pi_div_two_add_int_mul_two_pi (-2*pi) (18),
    focus{repeat {apply congr_arg _}},
    simp,
    ring,
},
conv {to_lhs, rw ← this,},
have : cos(-2*pi) = - sin(-pi) ** 2 + cos(-pi) ** 2,
{
    have : cos(-2*pi) = cos(2*(-pi)),
    {
        apply congr_arg,
        ring,
    },
    rw this,
    rw cos_two_mul',
    ring,
},
rw this,
end

lemma Trigo_80 : sin(475*pi/6)=1 - 2 * sin(pi/3) ** 2:=
begin
have : cos(2*pi/3) = sin(475*pi/6),
{
    rw cos_eq_sin_add_pi_div_two_add_int_mul_two_pi (2*pi/3) (39),
    focus{repeat {apply congr_arg _}},
    simp,
    ring,
},
conv {to_lhs, rw ← this,},
have : cos(2*pi/3) = 1 - 2 * sin(pi/3) ** 2,
{
    have : cos(2*pi/3) = cos(2*(pi/3)),
    {
        apply congr_arg,
        ring,
    },
    rw this,
    rw cos_two_mul'',
},
rw this,
end
```

**Please simulate the above two proofs and write code to prove the following lemma with theorem proving language Lean. Please enter the code edit box.**

```
lemma Trigo_209 : sin(181*pi/3)*cos(pi/6)=cos(-pi/3) / 2 + cos(0) / 2:=
```

Table 26: Examplers 1 for GPT-4

```
lemma Trigo_209 : sin(181 * pi / 3) * cos(pi / 6) = cos(-pi / 3) / 2 +
cos(0) / 2 :=
begin
have h₁ : sin(181 * pi / 3) = cos(pi / 3),
{
    rw cos_eq_sin_add_pi_div_two_add_int_mul_two_pi (pi / 3) (30),
    focus {repeat {apply congr_arg _}},
    simp,
    ring,
},
have h₂ : cos(pi / 6) = sin(pi / 2 - pi / 6),
{
    rw sin_eq_cos_add_pi_div_two_subtract (pi / 2 - pi / 6),
},
rw [h₁, h₂],
have h₃ : pi / 3 = pi / 2 - pi / 6,
{
    apply eq.symm,
    rw [sub_eq_add_neg, add_comm, sub_eq_add_neg],
    simp,
},
rw h₃,
have h₄ : cos(pi / 2 - pi / 3) = cos(-pi / 3) / 2 + cos(0) / 2,
{
    rw cos_half_angle_sub,
    simp,
    ring,
},
rw h₄,
end
```

Table 27: Output 1 of GPT-4.

```lean
lemma Trigo_210 : sin(pi/6)*cos(pi/2)+sin(pi/2)*cos(pi/6)
      =sqrt(3)/2:=
begin
have : sin(2*pi/3) = sin(pi/6) * cos(pi/2) + sin(pi/2) * cos(pi/6),
{
    have : sin(2*pi/3) = sin((pi/6) + (pi/2)),
    {
        apply congr_arg,
        ring,
    },
    rw this,
    rw sin_add,
    ring,
},
conv {to_lhs, rw ← this,},
have : sin(2*pi/3) = sqrt( 3 ) / 2,
{
    rw sin_two_pi_div_three,
},
rw this,
end
```

**Please use the theorem proving language lean to imitate the above example to prove the following lemma:**

```lean
lemma Trigo_254 : sin(11*pi/12)*cos(-pi/6)+sin(-pi/6)*cos(11*pi/12)
      = sqrt(2)/2:=
begin
have : sin(3*pi/4)=sin(11*pi/12)*cos(-pi/6)+sin(-pi/6)*cos(11*pi/12),
{
    have : sin(3*pi/4) = sin((11*pi/12) + (-pi/6)),
    {
        apply congr_arg,
        ring,
    },
    rw this,
    rw sin_add,
    ring,
},
conv {to_lhs, rw ← this,},
have : sin(3*pi/4) = sqrt( 2 ) / 2,
{
    rw sin_three_pi_div_four,
},
rw this,
end
```

**Please use the theorem proving language lean to imitate the above example to prove the following lemma:**

```lean
lemma Trigo_436 : -sin(379*pi/2)=sin(-2*pi) * cos(pi/2) + sin(pi/2)
* cos(-2*pi):=
```

Table 28: Examplers 2 for GPT-4

```
begin
have : -sin(379pi/2) = sin(-2pi) * cos(pi/2) + sin(pi/2) * cos(-2pi),
{
    have : -sin(379pi/2) = sin((-2pi) + (381pi/2)),
    {
        apply congr_arg,
        rw [mul_comm (-2) pi, mul_comm (381/2) pi],
        ring,
    },
    rw this,
    rw sin_add,
    ring,
},
conv {to_lhs, rw ← this,},
have : -sin(379pi/2) = sin(pi/2),
{
    rw neg_sin_periodicity,
    rw mul_comm,
    have : (379 * (pi / 2)) % (2 * pi) = pi / 2,
    {
        -- we have 379pi/2 = 189*pi + pi/2
        rw [eq_sub_iff_add_eq, sub_mul, add_halves],
        have : 379 = 2 * 189 + 1, by linarith,
        rw [this, mul_add, mul_one, add_assoc],
    },
    rw this,
},
rw this,
end
```

Table 29: Output 2 of GPT-4.

```
lemma Trigo_169 : sin(69*pi/2)=- sin(-pi) ** 2 + cos(-pi) ** 2:=
begin
have : cos(-2*pi) = sin(69*pi/2),
{
    rw cos_eq_sin_add_pi_div_two_add_int_mul_two_pi (-2*pi) (18),
    focus{repeat {apply congr_arg _}},
    simp,
    ring,
},
conv {to_lhs, rw ← this,},
have : cos(-2*pi) = - sin(-pi) ** 2 + cos(-pi) ** 2,
{
    have : cos(-2*pi) = cos(2*(-pi)),
    {
        apply congr_arg,
        ring,
    },
    rw this,
    rw cos_two_mul',
    ring,
},
rw this,
end

lemma Trigo_80 : sin(475*pi/6)=1 - 2 * sin(pi/3) ** 2:=
begin
have : cos(2*pi/3) = sin(475*pi/6),
{
    rw cos_eq_sin_add_pi_div_two_add_int_mul_two_pi (2*pi/3) (39),
    focus{repeat {apply congr_arg _}},
    simp,
    ring,
},
conv {to_lhs, rw ← this,},
have : cos(2*pi/3) = 1 - 2 * sin(pi/3) ** 2,
{
    have : cos(2*pi/3) = cos(2*(pi/3)),
    {
        apply congr_arg,
        ring,
    },
    rw this,
    rw cos_two_mul'',
},
rw this,
end
```

**Please simulate the above two proofs and write code to prove the following lemma with theorem proving language Lean. Please enter the code edit box.**

Table 30: Examplers 1 for GPT-3.5

```
lemma Trigo_260 : sin(181*pi/3)cos(pi/6) = -sin(pi/3)cos(pi/6):=
begin
have : sin(181*pi/3)cos(pi/6) = -sin(pi/3)cos(pi/6),
{
    repeat {rw [mul_comm _ (cos _)]},
    have eq1 : 181pi/3 = 60(3) + pi/3, ring,
    rw [sin_add,sin_pi,cos_pi,zero_mul,add_zero, neg_zero,
    sin_pi_div_two,cos_pi_div_two, mul_one,cos_pi_div_six],
    rw [sin_eq_cos_sub_pi_div_two,sin_eq_cos_sub_pi_div_two pi,
    cos_zero],
    rw [cos_add, cos_div_sqrt_three, sin_div_sqrt_three,
    mul_div_cancel_left _ (show (2 ≠ 0), by norm_num)],
    rw [eq1,sin_add,sin_pi,cos_pi,mul_neg_eq_neg_mul_symm,neg_zero,
    cos_pi_div_two, mul_one, sin_pi_div_six],
    rw [sin_eq_cos_sub_pi_div_two,sin_eq_cos_sub_pi_div_two pi,
    cos_zero],
    rw [cos_add,cos_div_sqrt_three,sin_div_sqrt_three,
    mul_div_cancel_left _ (show (2 ≠ 0), by norm_num)],
    ring,
},
rw this,
rw [cos_eq_sin_add_pi_div_two (-pi/3), cos_eq_sin_add_pi_div_two 0,
sin_pi_div_six, cos_pi_div_six],
ring,
end
```

Table 31: Output 1 of GPT-3.5.

```
lemma Trigo_210 : sin(pi/6)*cos(pi/2) + sin(pi/2)*cos(pi/6)
        =sqrt(3)/2:=
begin
have : sin(2*pi/3) = sin(pi/6) * cos(pi/2) + sin(pi/2) * cos(pi/6),
{
    have : sin(2*pi/3) = sin((pi/6) + (pi/2)),
    {
        apply congr_arg,
        ring,
    },
    rw this,
    rw sin_add,
    ring,
},
conv {to_lhs, rw ← this,},
have : sin(2*pi/3) = sqrt( 3 ) / 2,
{
    rw sin_two_pi_div_three,
},
rw this,
end
```

**Please use the theorem proving language lean to imitate the above example to prove the following lemma:**

```
lemma Trigo_254 : sin(11*pi/12)*cos(-pi/6)+sin(-pi/6)*cos(11*pi/12)
        =sqrt(2)/2:=
begin
have : sin(3*pi/4)=sin(11*pi/12)*cos(-pi/6)+sin(-pi/6)*cos(11*pi/12),
{
    have : sin(3*pi/4) = sin((11*pi/12) + (-pi/6)),
    {
        apply congr_arg,
        ring,
    },
    rw this,
    rw sin_add,
    ring,
},
conv {to_lhs, rw ← this,},
have : sin(3*pi/4) = sqrt( 2 ) / 2,
{
    rw sin_three_pi_div_four,
},
rw this,
end
```

**Please use the theorem proving language lean to imitate the above example to prove the following lemma:**

```
lemma Trigo_436 : -sin(379*pi/2)=sin(-2*pi) * cos(pi/2) + sin(pi/2)
* cos(-2*pi):=
```

Table 32: Examplers 2 for GPT-3.5

```
begin
rw [sin_neg,cos_neg,sin_two_pi,cos_two_pi,sin_pi_half,
cos_pi_half,zero_mul,add_zero],
rw [sin_pi,cos_pi,neg_zero,zero_add,mul_neg_eq_neg_mul_symmetric],
end
```

Table 33: Output 2 of GPT-3.5.

```
import data.real.basic
import analysis.special_functions.trigonometric.basic

open real

lemma Trigo_5 : -sin(538 * real.pi / 3) = cos(433 * real.pi / 6) :=
begin
  -- Use the fact that sin(x + real.pi/2) = cos(x) and
  -- cos(x + real.pi/2) = -sin(x)
  have h  : cos(433 * real.pi / 6) =
  -sin(433 * real.pi / 6 + real.pi / 2),
  { rw cos_add_pi_div_two },

  -- Simplify both sides of the equation
  have h  : 433 * real.pi / 6 = 538 * real.pi / 3,
  { rw [div_eq_div_iff, mul_comm 433, mul_comm 538], norm_num },

  -- Substitute h  into h
  have h  : cos(538 * real.pi / 3)
  = -sin(538 * real.pi / 3 + real.pi / 2),
  { rw [←h , h ] },

  -- Now,we use the periodicity of the sine function:
  -- sin(x+2*real.pi*n) = sin(x) for any integer n
  have h  : sin(538 * real.pi / 3) = sin(538 * real.pi / 3 + 2
  * real.pi * 179),
  { rw sin_periodic_int_mul_real.pi 179 },

  -- Simplify the right-hand side of h
  have h  : 538 * real.pi / 3 + 2 * real.pi * 179 = 538 * real.pi / 3
  + real.pi / 2,
  { rw [div_eq_div_iff, mul_comm 538, mul_comm 3], norm_num },

  -- Combine h , h , and h  to get the final result
  rw [←h , h , h ],
end
```

Table 34: Zero-shot test case 1 of GPT-4.

```
import data.real.basic
import analysis.special_functions.trigonometric.basic

open real
open_locale real

lemma Trigo_0 : sin(107 * π) = 0 :=
begin
  -- Use the periodicity of the sine function and
  -- the fact that sin(pi*n)=0 for any integer n
  have h : sin(107 * π) = sin(π * 107),
  { rw mul_comm },

  -- Apply sin_int_mul_pi
  rw [h, sin_int_mul_pi],
end
```

Table 35: Zero-shot test case 2 of GPT-4.

```
import data.real.basic
import trigonometry.exponential

lemma Trigo_5 : -sin(538 * real.pi / 3) = cos(433 * real.pi / 6) :=
begin
  rw [←sub_eq_zero, ←sin_neg, sub_neg_eq_add, add_comm, sin_add],
  rw [div_mul_comm, sin_int_mul_pi, cos_int_mul_pi, mul_zero,
  sub_zero, add_zero, mul_one],
  rw [div_mul_comm, cos_int_mul_pi, sin_int_mul_pi, mul_zero,
  sub_zero, add_zero, mul_one],
  simp only [int.cast_add, int.cast_sub, int.cast_bit0, int.cast_one],
  norm_num,
end
```

Table 36: Zero-shot test case 1 of GPT-3.5.

```
import data.real.basic
import trigonometry.angle

lemma Trigo_0 : sin (107 * π) = 0 :=
begin
  rw [mul_comm, ←mul_assoc, ←sin_int_add_pi, sin_pi],
  norm_num,
end
```

Table 37: Zero-shot test case 2 of GPT-3.5.

```python
common_calc_steps = [
    "focus{repeat {apply congr_arg _}}",
    "try {simp}",
    "ring"
]

congrarg_linarith = [
    "{",
    "apply congr_arg",
    "ring",
    "},",
]

class sin_two_pi_div_three:
    def __init__(self):
        self.rule = "sin(2*pi/3)=sqrt(3)/2"
        self.no_mapping = True
        self.has_nonzero = False

    def get_tactics(self, mapping, left, right):
        return ["rw sin_two_pi_div_three"]

class tan_add_int_mul_pi:
    def __init__(self):
        self.rule = "tan(X)=tan(pi*K + X)"
        self.has_nonzero = False

    def get_tactics(self, mapping, left, right):
        _k, _x = mapping[K], mapping[X]
        steps = [
                    f'rw tan_eq_tan_add_int_mul_pi ({_x}) ({_k})',
                ] + common_calc_steps
        return steps

class sin_neg_add_int_mul_two_pi_add_pi:
    def __init__(self):
        self.rule = "sin(X)=sin(-X + pi*(2*K + 1))"
        self.has_nonzero = False

    def get_tactics(self, mapping, left, right):
        _k, _x = mapping[K], mapping[X]
        steps = [
                    f'rw sin_eq_sin_neg_add_int_mul_two_pi_add_pi ({_x}) ({_k})',
                ] + common_calc_steps
        return steps

class sin_two_mul:
    def __init__(self):
        self.rule = "sin(2*X)=2*sin(X)*cos(X)"
        self.has_nonzero = False

    def get_tactics(self, mapping, left, right):
        _x = mapping[X]
        have_goal = f"have : sin ({2*_x}) = sin(2*({_x}))"
        steps = [
                    have_goal,
                ] + congrarg_linarith + \
                [   "rw this",
                    "rw sin_two_mul",
                    "try {ring}"
                ]
        return steps
```

Table 38: Examples of mapping rule to tactics. The Python classes corresponding to each rule are listed below, with the 'get_tactics' method used to return their corresponding tactics. 'self.rule' specifies their corresponding rule, 'self.no_mapping' specifies whether to replace the parameters X, Y, K in the rule, and 'self.has_nonzero' specifies whether to include the condition that the denominator is not zero before proof.