# OpenReview forum: "TRIGO: Benchmarking Formal Mathematical Proof Reduction for Generative Language Models"
_EMNLP/2023/Conference — EMNLP 2023 Main_

### Official Review · Reviewer_ET7v · 2023-08-04

**Soundness:** 4

**Excitement:**

4: Strong: This paper deepens the understanding of some phenomenon or lowers the barriers to an existing research direction.

**Paper Topic And Main Contributions:**

The paper explores in detail the problem of Trigonometric Expression Reduction (TRIGO) with Language Models, a task that can be used to develop and evaluate symbolic and numerical reasoning capabilities that are necessary for Automated Theorem Proving (ATP) and mathematical reasoning in general.

The paper proposes a new resource, named TRIGO, along with a data generation methodology that combines expert annotation with a formal environment (Lean). Specifically, TRIGO is composed of three subsets, TRIGO-web, TRIGO-real, and TRIGO-gen, that combine synthetic and real-world data to train and test models on problems of different complexity and investigate out-of-distribution generalisation.

The authors provide an in-depth evaluation and analysis of the performance of generative models, combining GPT2 with a proof search algorithm, and exploring the limitations of GPT4.

**Questions For The Authors:**

How do you ensure the quality of the annotated data and the agreement between annotators?

**Reasons To Accept:**

- The paper is very well-written and well motivated
- TRIGO represents an extensive resource that can support future research on the evaluation and improvement of the mathematical and symbolic reasoning capabilities of Language Models and Machine Learning approaches in general.
- The paper proposes an original methodology for data generation that combines human experts with a formal environment. The proposed methodology can serve as an inspiration for future work in the field.
- Extensive empirical evaluation on state-of-the-art models.

**Reasons To Reject:**

- I believe the paper would benefit from a deeper discussion on how the quality of the generated dataset is verified. For example, human annotation is involved, but I could not find any discussion or report of inter-annotator agreement and quality check.
- The evaluation is performed using a single family of generative models (i.e. GPT). As GPT4 is not freely available to the community and much of the details are unknown, I believe the experiments would benefit from an additional comparison between open models (for instance comparing GPT2 with T5). *Update*: This limitation is partially addressed in the authors's rebuttal with new results.

**Reproducibility:**

3: Could reproduce the results with some difficulty. The settings of parameters are underspecified or subjectively determined; the training/evaluation data are not widely available.

**Reviewer Confidence:**

4: Quite sure. I tried to check the important points carefully. It's unlikely, though conceivable, that I missed something that should affect my ratings.

---

> ### Author Rebuttal · Authors · 2023-08-29
>
> #### To Reviewer ET7v
>
> Thank you for your thoughtful comments and detailed feedback.
>
> **Question 1: I believe the paper would benefit from a deeper discussion on how the quality of the generated dataset is verified. For example, human annotation is involved, but I could not find any discussion or report of inter-annotator agreement and quality check. How do you ensure the quality of the annotated data and the agreement between annotators?**
>
> Response:
>
> We use Lean-Gym and Lean verifier to ensure the genearted and annoated proof is correct.  Specifically, for the generated proof, we only retain those proof that can reach a "no goals" state (indicating a correct proof) when interact with Lean-Gym. For annotated data, we submit them to Lean verifier and perserve the data that can pass the verification process.
>
> We consider a correct annotation also has relatively high qualtiy. Unlike some other tasks whose annotation is subjective and largely varies among annotators, the formal envrioment provides a rigorous and objective evaluation. If a annotation can pass the verifier, all annoator will agree the annoation is correct, leaving no room for ambiguity. Thus, we do not calculate the inter-annotator agreement (IAA) or observer consistency similar to other tasks, but use verifier to ensure the correctness of the annoated proof. We will include the above discusion in the revised version.
>
>
> **Question 2: The evaluation is performed using a single family of generative models (i.e. GPT). As GPT4 is not freely available to the community and much of the details are unknown, I believe the experiments would benefit from an additional comparison between open models (for instance comparing GPT2 with T5).**
>
> Response:
>
> Due to resource and time constraints, we only conducted experiments using T5-base on TG-real and TG-1, TG-2, TG-3. The detailed experimental settings are as follows: we set the maximum number of training epochs to 5, the initial learning rate to 5e-5, and used a linear learning rate scheduler. The batch size was set to 512, and we trained the model using 8 V100 GPUs. However, We found that T5-base only successfully completed a full proof on the simplest subdataset TG-1. Therefore, we conducted experiments and analysis on its single-step proof, and the experimental results are as follows:
>
> |        | TG-1  |  TG-2  |  TG-3  | TG-real  |
> |  ----  | ----  | ----  | ----  | ----  |
> | T5-base all-proof  | 74.1 | 69.1 | 65.0 | 48.7 |
> | GPT2-base all-proof  | 95.8 | 92.1 | 89.3 | 66.1 |
> | T5-base have  | 90.2 | 81.1 | 76.6 | 3.2 |
> | GPT2-base have  | 92.4 | 82.9 | 78.9 | 11.4 |
> | T5-base w/o have  | 69.5 | 65.3 | 61.3 | 60.3 |
> | GPT2-base w/o have  | 96.5 | 94.8 | 92.4 | 79.2 |
>
> The terms "all-proof," "have," and "w/o have" in the above table refer to the following: "all-proof" represents all single-step proofs filtered from TG-1, TG-2, TG-3, and TG-real datasets. "have" indicates single-step proofs that include the "have" tactic. "w/o have" refers to single-step proofs that do not include the "have" tactic. We conducted experiments to compare the performance of the gpt2-base and T5-base models on single-step proofs. We observed that they exhibited similar performance on the most difficult proofs, which involved stating the proof subgoals and included the "have" tactic. The primary performance difference between the two models was seen in the tactics that did not include the "have" tactic. On the program synthesis dataset, T5-base demonstrated an average accuracy that was 29.2% lower than gpt2-base, while on TG-real, its accuracy was lower by 18.9% compared to gpt2-base. The lower performance of T5-base on tactics that do not include the "have" tactics has led to difficulties in completing a full proof. We will include the above results and analysis in the revised version.

---

### Official Review · Reviewer_W12Q · 2023-08-04

**Soundness:** 3

**Excitement:**

4: Strong: This paper deepens the understanding of some phenomenon or lowers the barriers to an existing research direction.

**Paper Topic And Main Contributions:**

This paper contributes new data resources for the evaluation of formal mathematical proof reduction capabilities. The paper primarily describes the data curation process. It also presents some evaluations of GPT-2 and GPT-4.

Trigonometry problems are first sourced from Tiku, an open online repository of math problems across elementary, middle, and high school levels. First, a pool of annotators write out the step-by-step reduction process. Then each step is translated into the Lean formal language. Finally, additional problems are generated in Lean by using pre-existing rules, and substituting values (e.g. pi/2 for pi/3).


**Reasons To Accept:**

The paper contributes new data resources for mathematical proof reduction. Trigonometry seems to be a relatively underexplored area in proof datasets. The paper presents relatively thorough experiments to understand the gaps between the curated and generated datasets. The paper also assess the impact of different proof search algorithms, expert iteration, and an expanded set of angles.


**Reasons To Reject:**

Although it’s a well-curated dataset and it’s always important to contribute additional benchmarks, the specifics of the impact are not clear to me from reading the paper alone. Is the automatic proof generation the main methodological contribution? Is it possible to train on Trigo and have performance improve on other proof benchmarks?

Some of the steps of the dataset curation process could be described with more precision.
* Line 203: “To expand our dataset, we further collect additional trigonometry reduction problems from different websites.” It seems like only a small percentage of problems were sourced from outside Tiku. It would be great to specify the websites, the number from each website, and why additional sourcing was needed, since it seems that the total dataset size did not increase by much.
* Section 4.2 It would be great to show some examples, such as screenshots, of the annotation software and flow. It’s also not clear if the annotators during this step are the authors themselves, or if there were other annotators.
* It would be great to attach an appendix to Section 4.3 to clarify on which additional steps that PhD students had to fix. And again, it would be great to specify how they differ from the initial annotator pool.



**Reproducibility:**

4: Could mostly reproduce the results, but there may be some variation because of sample variance or minor variations in their interpretation of the protocol or method.

**Reviewer Confidence:**

2: Willing to defend my evaluation, but it is fairly likely that I missed some details, didn't understand some central points, or can't be sure about the novelty of the work.

**Typos Grammar Style And Presentation Improvements:**

Line 62: trigonometric
Line 139: Should citet instead of citep

---

> ### Author Rebuttal · Authors · 2023-08-29
>
> #### To Reviewer W12Q
>
> Thank you for your thoughtful comments and detailed feedback.
>
> **Question 1: Is the automatic proof generation the main methodological contribution?**
>
> Response:
>
> Our main contribution is the TRIGO benchmark, which is an evaluation framework for language models on formal mathematical reasoning with complex numerical inference. The benchmark construction involves automatic proof generation techniques to create training and testing sets spanning various difficulty levels. Throuhg the generation program, we can control the theorems sampling distribution, generate question with desried difficulty, and verify the correctness of the proofs automatically. Additionally, we also utilize automatic proof generation directly on TRIGO-real to expand the scale of the training dataset.
>
> **Question 2: Is it possible to train on Trigo and have performance improve on other proof benchmarks?**
>
> Response:
>
> Thank you for your valuable suggestion. Due to time and computational resource limits, we currently unable to include this experimental results here. However,  we are glad to evaluate whether the TRIGO sample can improve models’ performance on other benchmarks and include the results in the revised version if possible.
>
> **Question 3: It would be great to specify the websites, the number from each website, and why additional sourcing was needed, since it seems that the total dataset size did not increase by much.**
>
> Response:
>
> The questions collected from other websites refer to the websets found through search engines, including https://www.51jiaoxi.com/, https://wenku.baidu.com/, and https://www.gaokao.com/zyk/gzsj/. These sources contain high school math exam questions with standard answers. Throughout the collection process, we aimed to gather data randomly whenever possible, ensuring diversity in the distribution of the test set. The additional collection of TRIGO-web was conducted with the goal of obtaining a more extensive test set and a broader range of sources to reflect the model's performance on real human exam questions.
>
> **Question 4: It would be great to show some examples, such as screenshots, of the annotation software and flow. It’s also not clear if the annotators during this step are the authors themselves, or if there were other annotators.**
>
> Response:
>
> In Figure 7, we present screenshots of the annotation software and its workflow. The formalization process of trigonometric function proofs includes the authors, but the process of annotating the steps for trigonometric expressions reduction does not involve the authors.
> Specifically, our annotation team consists of four Master's students and three PhD students. The Master's students are responsible for completing the annotation of informal proof steps, while the Ph.D students focuse on formalizing the informal proofs into Lean. Four Master's students firstly simplify the trigonometric functions, then Ph.D students manually translate these informal proofs to Lean and submit them to the Lean theorem prover for correctness checking.
>
> **Question 5: It would be great to attach an appendix to Section 4.3 to clarify on which additional steps that PhD students had to fix. And again, it would be great to specify how they differ from the initial annotator pool.**
>
> Response:
>
> Thank you for your valuable suggestion. In the revised version, we will attach an appendix to Section 4.3 to provide clarification on the additional steps that Ph.D. students had to address. The initial annotator pool was tasked solely with annotating specific trigonometric expressions reduction processes. After annotating the reduction process, Ph.D. students translate the informal proof to formal proof with Lean.

---

### Official Review · Reviewer_Gc1T · 2023-08-04

**Soundness:** 3

**Excitement:**

4: Strong: This paper deepens the understanding of some phenomenon or lowers the barriers to an existing research direction.

**Paper Topic And Main Contributions:**

In this submission, the authors present TRIGO, a dataset of trigonometric expressions for training and studying the performance of LLMs on reducing these expressions. The authors describe their data gathering, data annotation (proof completion), and data generation process before providing experiments showing that training GPT-2 models on this dataset learn to reduce trigonometric expressions effectively (even compared to GPT-4, although this comparison is a bit unfair).


**Questions For The Authors:**

- Please elaborate more on the recruitment of the annotators and Ph.D. students completing the proofs. How were they selected, how much human effort was putting into this dataset?
- I browsed through the tiku website. Most of the questions were ‘fill in the blanks’ type of questions, how exactly was the data gathered, please elaborate
- What are the “different” websites on page 3, line 205?

**Reasons To Accept:**

- Studying mathematical reasoning capabilities of LLMs is an important research direction
- Useful datasets are rare, the authors contribute to the data scarcity problem
- I see the most value of this dataset in the manual annotation effort done by Ph.D. students (page 4, line 288+)
- Experimental results show that the dataset can be used to effectively train even small LLMs; promising results

**Reasons To Reject:**

- Some of the results are not surprising (with more human-level input, the model will improve in distribution)
- The dataset is limited in size, making it only usable for fine-tuning
- The comparison to GPT-4 is unfair, this needs to be more clearer in the paper (because despite the introduction/abstract, I doubt that GPT-4 has seen that much Lean)


**Reproducibility:**

2: Would be hard pressed to reproduce the results. The contribution depends on data that are simply not available outside the author's institution or consortium; not enough details are provided.

**Reviewer Confidence:**

3: Pretty sure, but there's a chance I missed something. Although I have a good feel for this area in general, I did not carefully check the paper's details, e.g., the math, experimental design, or novelty.

**Typos Grammar Style And Presentation Improvements:**

There are a few grammar issues, especially later in the paper, and in lines 68-87 on page 1. I would recommend running a checker.

---

> ### Author Rebuttal · Authors · 2023-08-29
>
> #### To Reviewer Gc1T:
> Thank you for your thoughtful comments and detailed feedback.
> **Question 1: The dataset is limited in size, making it only usable for fine-tuning.**
> Response:
>
> The dataset we have created is used for fine-tuning and testing the model's ability to perform mathematical reasoning involving complex combinations of numbers. We achieve this by manually collecting exam questions from the real world, annotating their simplification processes, and formalizing these processes. Additionally, we control the complexity of proof problem by manipulating their distributions, including numerical value ranges and proof difficulty. Lastly, we assess the model's generalization capabilities on both synthesized and real world problem. Due to resource constraints, we are unable to annotation large amouts of data for large scale pre-training.
>
> **Question 2: The comparison to GPT-4 is unfair, this needs to be more clearer in the paper (because despite the introduction/abstract, I doubt that GPT-4 has seen that much Lean).**
>
> Response:
>
> Firstly, we believe that Large Language Models (LLMs) such as GPT-4 have already shown the capability of processing formal language, particularly in the context of translating the informal proofs to formal proofs [1,2]. Furthermore, as indicated in reference [3], LLMs have substantial potential  in proving formal theorems with proof assistants like Lean. [3] demonstrated that GPT-4 can generate proofs accepted by Lean in a zero-shot manner, establishing GPT-4 as a strong baseline. These  results indicates the possibility of Lean example exists in the GPT-4 training corpus as numerous proofs were publicly accessible on GitHub prior to GPT-4's data cutoff date (September 2021) [4].
>
> The purpose of introducing GPT-4 is to explore the limits of language models in tasks involving complex numerical comprehension. For instance, as shown in Figure 2, we initially probed GPT-4's performance in completing entire proofs with human feedback errors. Subsequently, in Table 5, we delved into GPT-4's ability to complete single-step proofs. Furthermore, in Appendix L, we demonstrated GPT-4's capability to complete an entire proof in a single attempt when provided with a 2-shot complete proof. The above experiments have shown that gpt-4 is capable of generating partially correct proofs on TRIGO. However, it still exhibits common phenomena of language models such as hallucinations and inability to accurately provide equations. This suggests that our dataset poses a significant challenge for GPT-4.
>
> We will include the above analysis and clarification in the revised version to justify the reason of evaluating GPT-4 on our proposed TRIGO.
>
>
>
> **Question 3: Please elaborate more on the recruitment of the annotators and Ph.D. students completing the proofs. How were they selected, how much human effort was putting into this dataset?**
>
> Response:
>
> Our annotation team consists of four Master's students and three PhD students. The Master's students are responsible for completing the annotation of informal proof steps, while the Ph.D students focuse on formalizing the informal proofs into Lean. The four Master's students holds a Master's degree in Computer Science. They have received rigorous mathematical training and possess in-depth knowledge in the process of reducing trigonometric expressions. The three Ph.D students are in the field of computer science specializing in formal theorem proving. Four Master's students firstly simplify the trigonometric functions, then Ph.D students manually translate these informal proofs to Lean and submit them to the Lean theorem prover for correctness checking. The combined process of data annotation and mathematical formalization took one month to complete.
>
> **Question 4: I browsed through the tiku website. Most of the questions were ‘fill in the blanks’ type of questions, how exactly was the data gathered, please elaborate.**
>
> Response:
>
> We instructed the annotators to collect problems related to trigonometric expression reduction from the trigonometric section of the "tiku" website. For the fill-in-the-blank questions, we asked the annotators to gather the corresponding expressions and answers. Once collected, annotators manually filtered out questions whose answers did not match the expressions. If the collected questions lacked a reduction process (such as in the case of fill-in-the-blank questions), we required annotators to use an interactive annotation system to annotate the specific reduction steps.
>
> **Question 5: What are the “different” websites on page 3, line 205?**
>
> Response:
>
> We utilized search engines to gather questions from various websites, including  https://www.51jiaoxi.com/, https://wenku.baidu.com/, and also from high-school math exam papers provided at https://www.gaokao.com/zyk/gzsj/. Our intention was to avoid collecting questions solely from a single source, and  to obtain a diverse range of questions.
>
> >[1] Wu, Yuhuai, et al. "Autoformalization with large language models." Advances in Neural Information Processing Systems 35 (2022): 32353-32368.
>
> >[2] Jiang, Albert Q., et al. "Draft, sketch, and prove: Guiding formal theorem provers with informal proofs." arXiv preprint arXiv:2210.12283 (2022).
>
> >[3] Yang, Kaiyu, et al. "LeanDojo: Theorem Proving with Retrieval-Augmented Language Models." arXiv preprint arXiv:2306.15626 (2023).
>
> >[4] GPT-4 Technical Report.

---

### Meta-Review · Area_Chair_S8Ri · 2023-09-18

**Recommendation:** 4

**Metareview:**

This paper provides a benchmark measuring how well ML language models can reduce formal mathematical proofs.

Pros:
- Understanding mathematical reasoning capabilities of ML language models is a relevant task, even before the advent of ChatGPT, but even more important now that users commonly expect language models to assist with maths-related topics.
- The datasets, algorithms, and code are openly available and will be useful to the community.
- Well-written and well motivated.
- Proposes an original data-generation methodology combining human experts with a formal environment; can inspire future work.
- Extensive empirical evaluation on state-of-the-art models.

Cons:
- Somewhat small data set, perhaps restricting the use case to fine-tuning only.
- Manual annotation effort is not super clearly described, but additional details were provided in the review threads and these could be included into the paper.
- Some concerns around clarity and fairness of experiments involving GPT-4, which were addressed in the review threads and clarifications could be incorporated into the paper.

---

### Decision · Program_Chairs · 2023-10-07

**Decision:**

Accept-Main

**Comment:**

This paper provides a benchmark measuring how well ML language models can reduce formal mathematical proofs.

Pros:
- Understanding mathematical reasoning capabilities of ML language models is a relevant task, even before the advent of ChatGPT, but even more important now that users commonly expect language models to assist with maths-related topics.
- The datasets, algorithms, and code are openly available and will be useful to the community.
- Well-written and well motivated.
- Proposes an original data-generation methodology combining human experts with a formal environment; can inspire future work.
- Extensive empirical evaluation on state-of-the-art models.

Cons:
- Somewhat small data set, perhaps restricting the use case to fine-tuning only.
- Manual annotation effort is not super clearly described, but additional details were provided in the review threads and these could be included into the paper.
- Some concerns around clarity and fairness of experiments involving GPT-4, which were addressed in the review threads and clarifications could be incorporated into the paper.